# Development of Soil Moisture Profiles Through Coupled Microwave-Thermal Infrared Observations in the Southeastern United States

Vikalp Mishra[1,2], James F. Cruise[1], Christopher R. Hain[3], John R. Mecikalski[4], and Martha C. Anderson[5]

[1]Earth System Science Center, The University of Alabama in Huntsville, Huntsville, AL, USA
[2]NASA-SERVIR, Marshall Space Flight Center, Huntsville, AL, USA
[3]NASA Marshall Space Flight Center, Earth Science Branch, Huntsville, AL, USA
[4]Atmospheric Science Department, University of Alabama in Huntsville, Huntsville, AL, USA
[5]Hydrology and Remote Sensing Laboratory, USDA Agricultural Research Service, Beltsville, MD USA

**Correspondence:** Vikalp Mishra (vikalp.mishra@nasa.gov)

**Abstract.** The principle of maximum entropy (POME) can be used to develop vertical soil moisture (SM) profiles. The minimal inputs required by the POME model make it an excellent choice for remote sensing applications. Two of the major input requirements of the POME model are the surface boundary condition and profile-mean moisture content. Microwave-based SM estimates from Advanced Microwave Scanning Radiometer (AMSR-E) can supply the surface boundary condition whereas thermal infrared-based moisture estimated from the Atmospheric Land EXchange Inverse (ALEXI) surface energy balance model can provide the mean moisture condition. A disaggregation approach was followed to downscale coarse resolution ($\sim$25 km) microwave SM estimates to match the finer resolution ($\sim$5 km) thermal data. The study was conducted over multiple years (2006-2010) in the southeastern U.S. Disaggregated soil moisture estimates along with the developed profiles were compared with the Noah land surface model (LSM), as well as *in-situ* measurements from 10 Natural Resource Conservation Services (NRCS) Soil Climate Analysis Network (SCAN) sites spatially distributed within the study region. The overall disaggregation results at the SCAN sites indicated that in most cases disaggregation improved the temporal correlations with unbiased root mean square differences (ubRMSD) in the range of 0.01-0.09 $m^3 m^{-3}$. The profile results at SCAN sites showed a mean bias of 0.03 and 0.05 ($m^3 m^{-3}$); ubRMSD of 0.05 and 0.06 ($m^3 m^{-3}$); and correlation coefficient of 0.44 and 0.48 against SCAN observations and Noah LSM, respectively. Correlations were generally highest in agricultural areas where values in the 0.6-0.7 range were achieved.

## 1 Introduction

Although soil moisture (SM) represents a relatively small part of the overall hydrologic cycle, it is perhaps the most important part to human survival. SM is the source of water for all vegetation on Earth. It also plays an important role in water and energy exchanges between the land surface and atmosphere. Hydrologically, SM is an indicator of drought or lack thereof, and antecedent moisture conditions are important determinants of runoff response to rainfall events. Thus, SM is a vital part of any terrestrial ecosystem analysis as well as land surface and climate models.

Although SM can be measured *in-situ*, these observations are necessarily point based and may not be indicative of conditions over larger areas. In addition, *in-situ* SM data are only available generally at a few spatially sparse locations (Aghakouchak et al., 2015) though a number of field campaigns over the years have produced high-density observations globally, but only for very short time periods over limited domains. Thus, remote sensing is increasingly being relied upon to supply SM observa-

tions. Much of the recent efforts in remote sensing of SM estimation have been focused on surface or near surface observations (0-5 cm); however, moisture throughout the root zone can be just as important. The moisture within the root zone exerts a controlling influence on land-atmospheric fluxes of energy and water under vegetated condition. The actual distribution of root zone moisture is a function of vegetation canopy root density and distribution (Mishra et al., 2015). For this reason, SM at shallow depths (< 100 cm) is known to be extremely variable both as functions of time (Starks et al., 2003) and depth (Scott

et al., 2003).

Remotely sensed SM estimates, particularly from MW sensors are often assimilated into land surface and agricultural models in order to provide near real time updates of model states (Liu et al., 2011; Lievens et al., 2015; Pinnington et al., 2018; Ridler et al., 2014; Yang et al., 2016). However, such models often are not effective in assimilating surface SM into the deeper layers of the model (Kumar et al., 2009; Draper et al., 2011). This deficiency in model SM assimilation could be mitigated if the entire

SM profile was developed from remotely sensed data and then assimilated into the model. However, remote sensing alone cannot deduce the distribution of moisture within a soil column. The purpose of the study was to develop and test a method to merge surface and root zone SM estimates from remote sensing sources to develop an entire SM profile for assimilation into land surface models (LSM).

Although several approaches have been proposed for determining SM profiles, most require either observed profile data so

that a regression or inversion model can be developed (Arya and Richter, 1983; Kondratyev et al., 1977; Kostov and Jackson, 1993; Srivastava et al., 1997; Singh, 1988). Another common approach is to estimate surface or total root zone moisture using remote sensing and then assimilate those observations into a LSM to determine root zone SM distributions. The NASA Land Information System [LIS, (Kumar et al., 2006)] contains a suite of LSMs and data assimilation tools for this purpose that are commonly utilized as a source of SM data. However, as mentioned above, studies have shown that the LSM's are not

particularly effective in this regard and the models themselves have their own issues (e.g., bias, ancillary data requirements, etc).

Due to the inherent complexities involved with the movement of SM in the column, several studies have argued that SM uncertainties and complexities can be best described through the description of its entropy (Mays et al., 2002; Pachepsky et al., 2006; Singh, 2010). The maximization of entropy characterizes the diffusion of moisture through the soil column over a period

of time. The principle of maximum entropy (POME) states that if the inferences had to be drawn from incomplete information then they should be based on the probability distribution with maximum entropy allowed by the *a-priori* information. Al-Hamdan and Cruise (2010) used the maximum entropy formulation of Jaynes (Jaynes, 1957a, b) based on the Shannon entropy (Shannon, 1948) to formulate the POME-based SM profile development algorithm. Unlike other methods, the entropy approach suffers from no *a-priori* assumptions about the nature or shape of the moisture profiles in real space. The method is a statistical

approach and guarantees the minimum variance unbiased profile subject to the boundary and initial conditions specified. This

could be an improvement over other analytical methods that do presuppose a functional form of the SM distribution. Singh (2010) provides a full explanation of the theory of entropy of moisture movement in porous media.

Subsequent to its introduction the POME method has been adopted and extended by several authors e.g., Mishra et al. (2013, 2015); Pan et al. (2011); Singh (2010). Initial studies by Al-Hamdan and Cruise (2010) and Singh (2010) compared their results against experimental data under laboratory settings. However studies by Pan et al. (2011) and Mishra et al. (2013) involved application and validation of the POME model outside laboratory environment. Later, Mishra et al. (2015) provided extensive validation of the profiles developed using the POME approach against a U.S. Department of Agriculture Soil Climate Analysis Network (SCAN) site located in northern Alabama, as well as with a detailed physically based mathematical model of moisture movement in the soil profile.

The objective of this study is to develop SM profiles from remotely sensed data over the southeastern U.S without the aid of observed profile data. The approach utilizes both microwave (MW) data (to supply surface estimates) and thermal infrared (TIR) estimates (for total root zone moisture) within the POME profile methodology. The POME model requires only the upper and lower boundary conditions, as well as the mean moisture content, as input. The surface and mean moisture contents can be supplied by satellite estimates, whereas the lower boundary condition (∼100-200 cm) is often fairly stable (Mahmood and Hubbard, 2007) and can be parameterized or used to tie the remotely sensed profile to model climatology when assimilated into a LSM/crop model. Therefore, the POME model is ideal for the integration of remotely sensed data from multiple sensors such as MW and TIR to develop a unified SM profile.

The goal of this research is to provide SM profiles at operational or near operational (1-5 km) spatial resolutions to be consistent with other fine scale hydrologic and agricultural modelling work taking place in the study region (Mishra et al., 2013; McNider et al., 2015). Consequently, before the SM profiles can be calculated, the disparity in spatial resolution between the MW and TIR data must be resolved. MW data are available at much coarser spatial resolutions (25-40 km) than are TIR data (1-10 km). The approach selected here is to downscale (or disaggregate) the coarse MW data to the resolution of the TIR data. This is accomplished via an evaporative efficiency method proposed by Merlin et al. (2013, 2012, 2015).The spatial resolution selected is 4.7 km (∼5 km hereafter) that corresponds to the operational scale of the NWS Multisensor Stage IV precipitation product (Lin and Mitchell, 2005). This facilitates the future integration of the profiles into operational land surface, hydrologic, or agricultural models that are currently operating in the region. It is quite possible that these models could be improved through assimilation of observed SM profiles, especially in regions of the world where climate information is sparse.

As stated earlier, the overall objective of the study is to determine the efficacy of SM profiles developed directly from remotely sensed data, without the use of a LSM or ancillary data. The study consists of four parts: (a) a multiyear disaggregation of the coarse resolution MW surface SM to the 5-km spatial resolution; (b) calculation of SM profiles for each 5-km grid using the POME approach with the downscaled MW data serving as the surface boundary condition and TIR estimates providing mean SM; (c) validation of the SM profiles against a gridded LSM and *in-situ* data; and (d) error analyses including evaluation of downscaled MW surface SM estimates against LSM and *in-situ* data. Two independent data sources are used for comparison and validation purposes, ground observations from 10 available SCAN sites and gridded 3-km Noah LSM SM data aggregated to the 5-km spatial resolution.

## 2 Study Area and Data Sources

### 2.1 Study Area

The study area for this research is the southeastern U.S consisting of four states including Alabama, Georgia, Florida and South Carolina (Fig. 1). The region is home to a significant amount of current hydrologic and agricultural research activity where accurate SM modeling is a premium (McNider et al., 2011, 2015; Mishra et al., 2013). The southeastern U.S. represents a subtropical humid climate that typically has relatively hot and humid summers and precipitation that is generally evenly distributed throughout the year. The mean annual precipitation is 1250-1500 mm based on the 1981-2010 period. Mean annual temperature ranges from $14^oC$ in Northern Alabama to nearly $24^oC$ in southern Florida. The region is roughly 31% forested; 54% shrubs; 12% agricultural land and rest of the area is covered by urban (1.9%), savanna (1.8%), water etc. according to Moderate Resolution Infrared Spectroradiometer (MODIS) 2008 land cover data aggregated to 5-km spatial resolution. The majority of the soils (nearly 80%) at the surface are classified as sand with loamy sand and sandy loam, as determined from the Soil Information for Environmental Modeling and Ecosystem Management (Miller and White, 1998). These soils are known to have relatively low water holding capacity that can lead to great temporal variation in upper layer (1-10 cm) SM conditions and relatively frequent short-term droughts (1-4 week period) during growing seasons in various parts of the region (McNider et al., 2015). Although the study region is overwhelmingly represented by a forest and shrub landscape, vegetation that are known to adversely affect the accuracy of MW signals, the study does present an opportunity to evaluate the performance of the merged MW/TIR profiles in a challenging environment and may provide greater confidence in the robustness of the system. Further, the Southeastern U.S. is one of the more data rich regions of the world (climate and soils data) providing ample opportunity for calibration as well as validation of results.

### 2.2 Data Sources

#### 2.2.1 Microwave Surface SM

Over the past several years, much attention has been given to the use of MW sensors to measure surface SM remotely e.g. Soil Moisture Ocean Salinity (SMOS); the Advanced Microwave Scanning Radiometer for Earth Observation (AMSR-E); Soil Moisture Active Passive (SMAP); the Special Sensor Microwave Imager (SSM/I) etc. (Kerr et al., 2010; Entekhabi et al., 2010a; Njoku et al., 2003; Paloscia et al., 2001). The use of the MW band is the only remote sensing technique that is physically based as well as quantitative (Kondratyev et al., 1977; Schmugge et al., 1992). Furthermore, due to their all-weather and day/night capabilities, MW sensors are widely used globally and offer high temporal data availability. This study employs one of the more extensively used and validated MW based SM data sets from the AMSR-E mission operating in the X-band frequency from the National Snow and Ice Data Center (NSIDC) and employs the standard NASA retrieval algorithm (Njoku et al., 2003). The NSIDC is one of two AMSR-E data sets supported by NASA with the other being the Vrije Universiteit Amsterdam - Land Parameter Retrieval Model (VUA-LPRM) data. The LPRM uses a single dual polarized channel (X- or C- band) to deduce relationships between geophysical variables such as SM and vegetation characteristics and brightness temperatures (Cho et al.,

2015). Several studies such as Wagner et al. (2007); Draper et al. (2011); Jackson et al. (2010); Gruhier et al. (2008) etc. have compared the two data sets and the general conclusion has been that the VUA-LPRM algorithm may be slightly superior in terms of correlation to *in-situ* data, especially at lower latitudes and sparse vegetation (Brocca et al., 2011). However, Jackson et al. (2010) found that for the southeast U.S, the NASA retrieval algorithm outperformed the LPRM in terms of bias and root

mean square error (RMSE). Furthermore, as pointed out by Njoku et al. (2005) and Jackson et al. (2010) the effects of radio frequency interference (RFI) on C-band signals are more pronounced over countries such as the U.S. and Japan, therefore X-band retrievals are preferred over such regions. Hence X-band based standard NASA (or NSIDC) data set was selected for this study. The daily Level-3 AMSR-E SM X-band product (AELand3) (Njoku, 2004) from the ascending (1:30 pm local time) overpass was collected for this study. The ascending overpass was selected to be consistent with the timings of the

TIR retrievals, which are forced with morning and local noon skin temperatures obtained from the Geostationary Operational Environmental Satellite (GOES) Imager instrument. The Level-3 AMSR-E SM estimate is a 25-km gridded data product.

### 2.2.2 Thermal Infrared - ALEXI

Techniques to retrieve root-zone moisture that rely upon TIR data are inferred from surface energy fluxes typically retrieved at relatively high spatial resolutions. TIR-based evapotranspiration (ET) estimates are generally related to land surface tempera-

ture (LST) and vegetation cover fraction. Models such as the Surface Energy Balance System [SEBS: (Su, 2002)]; the Surface Energy Balance Algorithm for Land [SEBAL: (Bastiaanssen et al., 1998)]; and the Two Source Energy Balance [TSEB: (Norman et al., 1995)] exploit this relationship with varying degrees of complexities. A two-source based Atmospheric Land Exchange Inverse (ALEXI) (Anderson et al., 1997, 2007; Hain et al., 2011) model has been implemented over the continental U.S. and used as a source of surface energy fluxes (Anderson et al., 1997; Norman et al., 2003); ET (Anderson et al., 2011b, 2012);

SM (Hain et al., 2009; Mishra et al., 2013); and an Evaporative Stress Index (Anderson et al., 2011a, 2013). A continental-scale implementation of the ALEXI model was used in this study to estimate instantaneous energy fluxes. ALEXI fluxes are available at approximately 4.7 km ($0.04^o$) spatial resolution on a daily time-step since the year 2000 over the continental U.S., generated using 15-min resolution GOES 10.7 $\mu$m channel TIR data. ALEXI estimates of actual ET and SM are used in this study. A known drawback of TIR-based methods is that they are limited to cloud-free conditions.

### 2.2.3 *In-situ* Observations

The study area contains 25 operational U.S. Department of Agriculture SCAN (Schaefer et al., 2007) monitoring stations. In addition to meteorological observations such as precipitation, air temperature, relative humidity etc. these monitoring stations measure soil temperature and moisture content primarily at depths of 5, 10, 20, 50 and 100 cm at hourly and daily time steps. The SCAN sites use Hydra Probes (Stevens) to observe SM conditions (Schaefer et al., 2007). Most of these

25 SCAN sites are located in northern and central Alabama. Ten sites with the most consistent data availability and with good geographical distribution across the study area were employed for the comparison. The SM data were obtained from http://www.wcc.nrcs.usda.gov/scan/. Table 1 lists the major land cover type (at 5-km scale) along with soil characteristics at the selected sites.

### 2.2.4 Noah Soil Moisture

The Noah SM product generated within the NASA LIS (Kumar et al., 2006) framework was selected as a complementary evaluation dataset. The Noah model SM product used in this study is provided by the NASA Short-Term Prediction Research and Transition Center (SPoRT). The model is driven by actual meteorological forcings from the North American Land Data Assimilation system-Phase 2 (NLDAS2) (Xia et al., 2012), and thus serves as a valuable comparison dataset by which to measure the MW downscaling and profile results. While Noah SM also has biases and uncertainties, the comparisons reveal regional patterns of agreement (disagreement) with the remote sensing estimates. In the event that the POME profiles prove to be superior to the LSM in certain instances, this would indicate that the LSM (or other hydrologic or agricultural models) might be improved through assimilation of the remotely sensed SM profiles. The comparison assumes that errors in the Noah model are independent from the errors associated with MW and TIR based estimates. Noah SM estimates are available in four layers: 0-10; 10-40; 40-100 and 100-200 cm depths. It should be noted that there are inconsistencies in the surface layer depths between Noah and MW data: The surface layer in the Noah model is the top 0-10 cm of the soil column, while the downscaled MW represents the top 0-2 cm. The 3-km Noah SM products were aggregated to 5-km scale to be consistent with the TIR products.

Additionally, the NLDAS2 gridded temperature forcing data ($0.125^o$ resolution) were also utilized for computing potential ET (PET). The NLDAS2 forcing data was available from NASA land data assimilation portal(https://ldas.gsfc.nasa.gov/nldas/ NLDAS2forcing.php). The GTOPO30 digital elevation model (DEM) was used as source of elevation information for the study area. The GTOPO30 product was made available by the U.S. Geological Survey's EROS Data Center (https://lta.cr.usgs.gov/GTOP30). The 1-km gridded soil characteristic data for the study area was available from the Soil Information for Environmental Modeling and Ecosystem Management (Miller and White, 1998).

## 3 Methodology

### 3.1 ALEXI Retrievals

#### 3.1.1 Surface Evaporation

A time differential application of the ALEXI model was performed to monitor the rise in LST from morning to local noon. The early rise in LST is used to diagnose the partitioning of net radiation into sensible, latent and soil heat fluxes. The rise in LST from morning to near-noon is known to be correlated with the moisture content of the soil: compared to a dry land surface, wetter surfaces warm slowly, thus requiring more energy for evaporating surface moisture (Hain et al., 2011; Kustas et al., 2001). The soil heat conduction flux is parameterized as a function of net radiation following (Santanello and Friedl, 2003); latent heat from the canopy (transpiration) is estimated assuming a non-stressed modified Priestley-Taylor (Priestley and Taylor, 1972) approach. Finally, the soil (surface) latent heat is the residual of the canopy latent heat and latent heat of the soil and canopy system $LE_s = LE_{sys} - LE_c$. Here $LE_s$, $LE_{sys}$ and $LE_c$ represent the latent energy of surface, system and

canopy, respectively. Detailed model description and derivation is provided in earlier studies (Anderson et al., 2007; Hain et al., 2011). If the residual is negative [an indicator of condensation, an unlikely process during daytime (Hain et al., 2011)] then the canopy transpiration is relaxed iteratively until it reaches zero. The surface evaporation from ALEXI is used to compute the soil evaporative efficiency (SEE) function required for the disaggregation (described in section 3.2).

### 5    3.1.2    Mean Root Zone Moisture Retrieval

The ratio of actual to potential ET ($f_{PET}$) is functionally related to the fraction of available water ($f_{AW}$). Multiple relationships between the ratios of PET and available water have been proposed with varying degrees of success including linear; non-linear; piecewise linear or threshold (Hain et al., 2009). Large-scale applications prefer simpler linear functions as sensitivity to SM is constant and thus relatively less detailed soil characteristics are required (Song et al., 2000). In this study a linear relationship
proposed by Wetzel and Chang (1987) is employed: $f_{PET} = 0.85 * f_{AW}$. The resulting ALEXI SM estimation is given as:

$$\theta_{ALEXI} = (\theta_{fc} - \theta_{wp})(0.85 * f_{AW}) + \theta_{wp} \tag{1}$$

Here $\theta_{fc}$ and $\theta_{wp}$ represent the field capacity and wilting point of the soil, respectively. ALEXI retrievals can be interpreted based on fraction of vegetation cover ($f_c$) as either surface moisture content ($f_c < 0.3$); predominantly root-zone moisture ($f_c > 0.75$) or a composite of both surface and root-zone moisture for $f_c$ between these limits. In this study Priestly-Taylor
PET was used with ALEXI actual ET to compute $f_{AW}$. It is argued that the SM retrieval from diagnosed evaporative fluxes is reasonable when the SM content is within the limits of wilting point and field capacity (Hain et al., 2011). Since in this study we are concerned with root zone SM content, it is not anticipated that the limitation mentioned will lead to significant error as conditions resulting in entire soil column through the root zone be above field capacity (or less than wilting point) would be rare.

### 20    3.2    Surface Disaggregation

The spatial resolution of the TIR- based ALEXI SM estimates are roughly 5x5 km. Thus, in order to utilize them in conjunction with the AMSR-E MW data, the coarse resolution MW surface estimates must be downscaled to match the ALEXI spatial scale. A physically based, semi-empirical SEE model in combination with a first order Taylor series expansion around the coarse resolution SM was used to map surface evaporative fluxes to SM content at finer resolutions. The SEE disaggregation approach
has become very popular recently and has been employed by several investigators at varying spatial scales and locations such as: Chen et al. (2017) disaggregated SMAP SM to 250-m [correlation ($r$): -0.3-0.72, RMSE: 0.06-0.27]; Malbéteau et al. (2016) used the algorithm to downscale coarse scale SM to 1-km resolution [r: 0.70-0.94, RMSD: 0.07-0.09]; Merlin et al. (2015) downscaled SMOS SM to 1-km scale [$r$: -0.22-0.64, root mean square difference (RMSD):0.05-0.32]; Molero et al. (2016) also downscaled coarse resolution SMOS SM to 1-km scale [$r$: 0.35-0.47, unbised RMSD (ubRMSD):0.04-0.12]. In
general, the disaggregation improves agreement with *in-situ* observations in comparison with coarse-scale estimates.

The disaggregation approach decouples the soil evaporation from the top few centimeters of the soil and the vegetation transpiration through ET partitioning. The disaggregation algorithm used in this study follows the concept of the DISaggregation

based on Physical and Theoretical scale CHange [DISPATCH: (Merlin et al., 2012, 2013, 2010)] model. The model accounts for aerodynamic resistance over bare soil in addition to soil parameters such as field capacity via the SEE. Detailed DISPATCH algorithm derivation and description is presented by Merlin et al. (2012). Here we represent the prominent disaggregation equation as:

$$SM_{HR} = SM_{LR} + \frac{\partial SM_{mod}}{\partial SEE} \big[ SEE_{HR} - \langle SEE_{HR} \rangle_{LR} \big] \tag{2}$$

HR and LR refer to the high and low-resolution variables, respectively. There have been multiple linear and non-linear relationships proposed between SEE and surface SM in the past (Budyko, 1961; Komatsu, 2003; Lee and Pielke, 1992; Manabe, 1969; Noilhan and Planton, 1989). A recent study by Djamai et al. (2015) suggest that the non-linear model performs better in humid climatic condition. Therefore non-linear model suggested by Noilhan and Planton was used in this study to guide the DISPATCH algorithm:

$$\frac{\partial SM_{mod}}{\partial SEE} = \frac{SM_{LR}}{cos^{-1}(1 - 2SEE_{LR}\sqrt{SEE_{LR}(1 - SEE_{LR})})} \tag{3}$$

### 3.2.1 Modified SEE Computation

The SEE, which can be defined as the ratio of actual to potential surface soil evaporation (Fang and Lakshmi, 2014; Merlin et al., 2010), is computed at the high resolution first, and then the SEE results are aggregated to the respective low resolution 25-km MW scale. The studies by Merlin et al. (2010, 2012) demonstrated the use of MODIS LST, Normalized Difference Vegetation Index (NDVI) and albedo to determine surface and vegetation temperature and evaporation. The SEE was defined as: $= \frac{T_{s,max} - T_{s,HR}}{T_{s,max} - T_{s,min}}$, where $T_{s,max}$ is the soil temperature at SEE = 0; $T_{s,min}$ is soil temperature at SEE = 1, and $T_{s,HR}$ represents soil temperature at the high resolution grid scale.

However, in this study we employed the ratio of the estimated surface evaporation from ALEXI to the potential evaporation to compute SEE directly at the 5-km ALEXI resolution. As mentioned earlier, the two-source land surface representation in ALEXI separates surface evaporation and canopy transpiration. The potential surface evaporation is calculated using the Hamon PET (Hamon, 1963). Hamon PET estimates are completely dependent upon atmospheric demand irrespective of soil and vegetation characteristics and can act as a proxy of potential surface evaporation (PE). This represents a subtle change in the definition of SEE from the Merlin formulation in that in our case all land cover/soil matrix combinations are weighted equally as opposed to being weighted by their assumed PE value as in Merlin (approximated as function of surface temperature). Since the Southeastern U.S. is an energy limited, water rich environment (Ellenburg et al., 2016), evaporation is controlled primarily by water availability and atmospheric demand; therefore, the effects of this change are not expected to be large. Hamon PET estimates have been found to be comparable to radiation based methods (e.g., Priestly-Taylor) to observed ET in the Southeastern U.S. at monthly or longer time scales (Lu et al., 2005), and are computed using air temperatures from the NLDAS2 forcing data subject to terrain adjustment. Terrain adjustment of coarse resolution temperature data was performed

using a 30 m digital elevation map of the region and constant lapse rate of -6.5 $K.km^{-1}$ (Cosgrove, 2003).Recently, Mishra et al. (2018) utilized the DISPATCH algorithm with modified SEE computation to disaggregate the coarse resolution SMAP SM over continental U.S. to 9 and 3-km resolutions with mean correlation of 0.47 and ubRMSD of 0.064 ($m^3m^{-3}$) against SCAN observations.

## 3.3   Profile Development

A multi-year vertical SM profile was developed for each ALEXI grid cell using the POME model developed by Al-Hamdan and Cruise (2010) over the study area. Maximizing the Shannon entropy, they developed a model to create monotonic profiles using boundary conditions and mean moisture content information via the method of Lagrange multipliers in effective soil moisture ($\Theta$) term:

$$\Theta(z) = \frac{ln[exp(\lambda_2\Theta_0) \pm exp(1-\lambda_1)(\frac{z}{L})]}{\lambda_2} \tag{4}$$

The Lagrange multipliers $(\lambda's)$ can be determined from application of the constraints and boundary conditions (surface effective saturation,$\Theta_0$) and mean effective saturation value of the soil column ($\overline{\Theta}$), $z$ is calculation depth, and L is total depth of the column. Eq. (4) is a monotonically increasing (+ sign) or decreasing (- sign) function, representing dry (increasing from the top boundary) and wet (increasing from the bottom boundary) case profiles. For details of the POME model please see the appendix A. Effective saturation values are mapped back to volumetric water content using soil water characteristics from Rawls et al. (1982) (see appendix B).

### *Handling Dynamic Cases:*

Experience has shown that not all SM profiles are monotonic as given by Eq. (4). In fact, it is clear that some profiles can be parabolic in shape (i.e., demonstrate an inflection point), especially immediately subsequent to rain events (dynamic case), or due to sharp changes in soil characteristics (Al-Hamdan and Cruise, 2010; Mishra et al., 2015) (see Fig. A1). Such cases are identified when mass balance cannot be kept by the monotonic assumption and thus Eq. (4) has no solution. In these cases, it is assumed that the inflection point is located in the soil layer with the greatest field capacity (Mishra et al., 2015). The POME model is then applied twice; from the surface to the inflection point, and then from the inflection point to the bottom boundary. This procedure was only required in 9% of the profiles generated in the study.

As mentioned previously, POME is a statistical approach that begins with the assumption of a uniform SM distribution initially and envisions the profile as either monotonically increasing or decreasing, or possessing one prominent inflection point. The final profile will then be the optimal one subject to the given boundary and initial conditions. Of course, this is a simplification of the true behavior of SM in the field; however, numerous previous studies have demonstrated that if given accurate input data, the error in the profiles will generally be less than < 3% (Al-Hamdan and Cruise, 2010; Singh, 2010; Mishra et al., 2015).

In the present application, the surface boundary condition is supplied by the downscaled AMSR-E estimates, the mean moisture content from the ALEXI model, and the lower boundary is parameterized as 50% of available water content. Experience has shown that SM behavior becomes less dynamic deeper into the soil column and thus selection of a constant value (in terms of effective moisture content) might serve as a useful first approximation. Since in many ways, this is a proof-of-concept study,
it was felt that this would be appropriate and would remain in the spirit of the Maximum Entropy concept that a minimum of *a-priori* information should be used in the calculation of the profiles. Incidentally this assumption serves to introduce error into the analysis as all profiles do not exhibit this behavior (see Fig. 5) (particularly in terms of volumetric SM content which is dependent on underlying soil characteristics) and even those that do may not stabilize around the particular SM state selected.

## 3.4 Temporal Compositing

The ALEXI data are available from 2000 to present and AMSR-E from 2002-2011. For this study, the years 2006-2010 were selected for analysis as the NRCS SCAN data was most consistently available during this period (nearly 92%). The ascending AMSR-E SM estimates were available 64.5% of the days on an average for all scan site locations while ALEXI retrievals were available on only 36% of the days due to cloud cover limitations. Earlier studies such as by Leng et al. (2017a, b) explored a vegetation and aerodynamic coefficient based gap filling algorithm for satellite derived SM estimations. Despite showing
promise, the proposed algorithm requires ancillary datasets that are not part of this study (e.g. wind speed) which might add additional error to the analysis. Whereas, temporal compositing can be used for gap filling, it further tends to reduce the day-to-day noise associated with the satellite retrievals (Anderson et al., 2011a). Further, there is a strong correlation between surface SM and moisture dynamics at lower layers for temporal lags of less than 5-days (Penna et al., 2013; Alfieri et al., 2017). Therefore, a three day moving window un-weighted mean was used on AMSR-E and ALEXI retrievals to develop a composite
dataset. Compositing of the ALEXI surface ET increased the mean data availability from 36 to nearly 63% over all scan sites and in the case of AMSR-E compositing ensured close to 100% data availability. The availability of pixels with intersection of AMSR-E and ALEXI data more than doubled from 22.5% to 58.7% for the study period over all sites.

## 3.5 Evaluation Metrics

The remote sensing derived SM profiles developed using the POME model were compared and validated against *in-situ* ob-
servations from 10 NRCS SCAN sites along with the gridded Noah LSM SM products over the study area. The data gaps in all three datasets restrict the possibility of time series analysis; therefore, pair-wise temporal statistical comparisons were performed using traditional matrices such as correlation coefficient ($r$), RMSD and bias. It has been argued that in cases with either the model or reference dataset being biased in mean or amplitude of fluctuations, the traditional RMSD tends to be an overestimation of true unbiased data (Entekhabi et al., 2010b). Therefore an ubRMSD in addition to traditional RMSD was
also computed. The ubRMSD can easily be computed by removing the bias term form the definition as:

$$ubRMSD = \sqrt{(RMSD^2 - Bias^2)} \tag{5}$$

To assess the quantitative error between three datasets against an unknown true observation, the triple collocation (TC) error estimation method was employed (Stoffelen, 1998). TC has become a very popular technique for simultaneous error analysis of three data sets since its adaptation to SM states by Scipal et al. (2008). The procedure is based on the assumption of linear relationships between the three estimates of the SM at a specific location and the unknown true value. The unknown truth is

eliminated from the linear error equations through subtraction and then cross multiplication to determine the error variances of the datasets relative to each other (Gruber et al., 2016). The assumption is that the errors in the three datasets are independent and random. Multiple recent studies have used the triple collocation method for error estimation [such as Crow et al. (2015); Yilmaz et al. (2014); Su et al. (2014); McColl et al. (2014) etc.]. A detailed review of method derivations and application to SM error estimation and analysis is presented by Gruber et al. (2016). In this study, the TC errors were computed using variances

and covariances of the datasets (McColl et al., 2014; Stoffelen, 1998; Su et al., 2014). The covariance based approach enables us to compute unscaled error variances directly. Further, the covariance notation also computes the sensitivity of the datasets $(\beta_i^2 \sigma_\Theta^2)$ against true SM signal. Here $\beta$ is the scaling parameter, $i$ is the dataset and $\sigma_\Theta^2$ is the variance of the true jointly observed SM signal. Sensitivity estimates can be used to further validate and inter-compare the datasets. Recently, McColl et al. (2014) proposed a methodology to compute the correlation coefficient between the involved datasets (referred as X,Y,Z)

and underlying true SM signal as:

$$R_i^2 = \frac{\beta_i^2 \sigma_\Theta^2}{\beta_i^2 \sigma_\Theta^2 + \beta_i^2 \sigma_{\epsilon_i}^2} \tag{6}$$

Here $\sigma_{\epsilon_i}^2$ is the error variance of the dataset where $i \in$ [X,Y,Z]. The sensitivity of individual dataset is estimated as:

$$\beta_X^2 \sigma_\Theta^2 = \frac{\sigma_{XY} \sigma_{XZ}}{\sigma_{YZ}} \qquad \beta_Y^2 \sigma_\Theta^2 = \frac{\sigma_{YX} \sigma_{YZ}}{\sigma_{XZ}} \qquad \beta_Z^2 \sigma_\Theta^2 = \frac{\sigma_{ZX} \sigma_{ZY}}{\sigma_{XY}} \tag{7}$$

## 4    Results and Discussions

### 4.1    Comparison with Noah LSM

For comparing SM profiles, 0-100 cm POME based profiles at 5 cm layer depth intervals were aggregated using the unweighted simple mean to the depths consistent with the Noah LSM: 0-10; 10-40; and 40-100 cm. The analysis can be approached from three perspectives: the surface values represent the MW downscaling; the bias represents the ALEXI model performance as it is providing the total SM content in the root zone; and the RMSD is representative of the entropy model since it measures the

moisture distribution within the soil column.

Figure 2 shows the statistics of multi-year temporal SM profile comparisons between the POME and the Noah LSM for the study region. The figure shows that the mean RMSD and ubRMSD tend to be relatively stable with depth over the entire region, an indication of relative stability for the profile developed using the POME model. As depth increased, pixel bias ranged from 0.05-0.13 m$^3$m$^{-3}$ indicating that the mean SM data from the ALEXI model is positively biased compared to the Noah LSM,

although the mean bias was $\leq 0.05$ m$^3$m$^{-3}$ for all layers. The overall RMSD at all layers was found to be under 0.085 in volumetric SM. Moreover more than 97% pixels across the study area showed ubRMSD of less than 0.06 m$^3$m$^{-3}$ across all layers, indicating good agreement between the POME model and the Noah SM estimates (Jackson et al., 2010). Comparing Fig. 2 with the landcover map (Fig. 1), it seems that the higher correlations ($r > 0.6$) occur more prominently in the agricultural dominant portions of the study area for the top two layers (0-40 cm). The overall correlations in the range of 0.46-0.54 across layer depths suggest that the temporal variabilities from remotely sensed driven POME model compared fairly well against Noah SM.

Comparison between POME and Noah SM profiles by land cover type (Fig. 3) indicate that the absolute bias tends to increase with depth in the savannah, shrub, and forest land covers while the reverse is evident for the urban, grass and crop coverages. It appears that overall bias is lowest in the savannah, forest, and agricultural land classes and since those classes (particularly forest) dominate the region, this naturally leads the relatively low overall region-wide bias shown in Fig. 2.

The RMSD (and ubRMSD) present an opportunity to judge the overall profile development process. It is clear from Fig. 3 that the RMSD improves from the surface to the middle layer and then increases again in the bottom layer in every land cover class except shrub. The top and bottom layer RMSD is being impacted by the boundary conditions placed on the POME integral by the MW and the parameterized lower boundary. Clearly, the POME process tends to improve the imprecise surface boundary as depth increases until the assumed lower boundary condition is encountered and results in deterioration of the profile RMSD.

In terms of correlation, the mid layer (10-40 cm) has the highest correlation (overall mean $r = 0.54$) for all land cover types with the highest mean correlation of 0.7 for crop dominated landcover. This further demonstrates the capabilities of the ALEXI model to estimate root-zone mean SM content in comparison to the Noah LSM. Incidentally, for most crops, the majority of the root mass is distributed in the top 60 cm of the soils column (Wu et al., 1999). The higher root density ensures the strong coupling of the land-plant-atmosphere system which tends to improve the accuracy of ALEXI in that zone. Increased correlations in the 10-40 cm layer indicate the ability of ALEXI to mimic the temporal patterns in the root-zone consistently relative to Noah. As depth increases, the root density is reduced and thus the coupling between land and atmosphere is also reduced. This fact, along with the relatively coarse parameterization of the lower boundary on the POME profile, leads to a relative decrease in correlation at layer 3 (40-100 cm) at all land covers except for trees (forest). The cropland showed the highest correlations with the Noah profile while keeping the RMSD and bias consistent with other land types. Agricultural areas demonstrated correlations ranging from 0.5 to 0.7 with a mean correlation of 0.62.

The overall analysis by layer depths appear to indicate that the profiles developed through the POME model using the disaggregated MW and the ALEXI derived mean SM content is in good agreement with the Noah LSM in the southeastern U. S. and in very good agreement in agricultural areas of the region.

## 4.2 Comparison with *in-situ* Observations

The comparison against Noah LSM SM estimates provided useful insights towards the performance of TIR-based SM profiles developed through the POME model. The comparisons against the LSM specifically adds to the analysis of results as a function

of land cover, yet as mentioned earlier, the analysis does not assume that Noah is a perfect model - it may have its own errors. Therefore multiple NRCS SCAN site *in-situ* observations are used for further validations. When comparing remotely sensed data to site specific *in-situ* observations, disparities in spatial scale and sensing depth must be considered. Although some authors prefer to remove bias due to the differing scales before comparisons are made (Brocca et al., 2011), it is also quite
common to do the comparisons without adjusting for scale, even when only one *in-situ* site is available (McCabe et al., 2005; Sahoo et al., 2008). In this study no bias corrections were performed.

Figure 1 shows the location of each of the sites used for validation along with the underlying land cover map. Table 1 summarizes the SCAN site characteristics, dominant land cover types and soil characteristics at the surface and 100 cm depth. Dominant land cover for sites 2009, 2114 and 2115 are predominantly savannas and forest type (hereafter referred as forest
sites), whereas sites 2013, 2037, 2038 and 2113 are a mix of cropland either with savannas or shrubs (hereafter referred as mixed cropland sites). Only sites 2027, 2078 and 2053 (hereafter referred as cropland sites) are predominantly cropland at the 5-km spatial resolution footprint. The crop and mixed crop sites are shown in bold in the following text. The SCAN sites monitored SM at depths of 5, 10, 20, 50 and 100 cm. The POME based profiles are developed at 5 cm layer depth increments down to 100 cm depth.

The results of the developed profiles in comparison to the SCAN site observations are shown in Fig. 4. First, it is evident in all the statistics except the correlation that the pattern demonstrated in the previous comparisons persists in that the statistics often tend to improve with depth with occasional deterioration when the lower boundary is encountered. Considering the performance of ALEXI initially, the bias appears reasonable in most cases where the majority of instances the absolute bias is less than 0.1 ($m^3m^{-3}$), but it appears to be best in the mixed cropland areas (mean absolute bias of 0.07 $m^3m^{-3}$ across
all depths) and worse in forested sites (mean absolute of 0.13 $m^3m^{-3}$). In fact, at seven of the ten total sites the overall bias is considerably less than the average moisture content at the SCAN site itself. At the two sites with the highest bias (2009 and **2027**), the mean moisture content from ALEXI was about twice the observations at all layers, indicating that the satellite estimates showed considerable positive bias (mean bias 0.17 and 0.13 $m^3m^{-3}$, respectively). Hain et al. (2011) pointed out that sensitivity of the ALEXI model decreases as moisture content nears either the wilting point or the field capacity. Both
sites 2009 and **2027** had sandy soils at the SCAN site and exhibited the lowest mean moisture content of all sites. At site 2009 with sandy soil through the column, the mean SM content was 0.05 $m^3m^{-3}$ against the wilting point of 0.033 $m^3m^{-3}$ while **2027** site had sand at the surface and sandy loam (wilting point = 0.095 $m^3m^{-3}$) at the 100 cm depth and the mean SM content was 0.12 $m^3m^{-3}$. Moreover, the site 2009 is located in a forest-dominated region. Whereas for site **2027** (located in southwest Georgia), the higher bias in remotely sensed observations can be attributed to additional SM content due to irrigation.
Southwest Georgia is one of the most irrigated regions of the study area. In contrast, the SCAN site observations are primarily governed by precipitation alone.

In the case of RMSD, half the sites showed an average RMSD of 0.1 $m^3m^{-3}$ or less. RMSD tends to be better at the mixed land use sites, while poor performances at sites 2009, 2115 and **2027** skewed the forest and cropland results respectively. As in the bias case, these sites demonstrated the highest mean RMSD values (Fig. 4). However, with the exception of these sites, the
average RMSD was less than the SCAN average moisture content in all cases. The ubRMSD, on the other hand, at all sites was

better with the overall ubRMSD for all layer depths and land cover types exhibiting an average ubRMSD of 0.07 m$^3$m$^{-3}$. The ubRMSD tended to improve with depth for all cases (Fig. 4) up to the depth of 50 cm, but showed a rise at the 100 cm depth as discussed previously. Improvements in ubRMSD with depth indicate the ability of the POME model to converge and correct itself from the effects of the noisy surface boundary condition until the assumed lower boundary affected the performance in

that layer.

The $r$ results are interesting and do not necessarily track the other two indices. It is clear from Fig. 4 that POME tended to perform better in agricultural land use areas than in other environments. Similar to the bias results, correlation was poorest at forested locations. In all, three sites showed average correlation above 0.5 with four other sites showing a correlation above 0.4. Two sites (2009, **2113**) produced average correlations of 0.16 and 0.32 across all depths. As discussed earlier, site 2009 is

forested while **2113** is located near a water body (Lake Catoma). Overall, the crop sites showed the highest correlations (0.51) followed by mixed crop sites (0.42), an indication of the ability of the satellite derived surface and mean moisture content estimates to mimic wetting and drying patterns over time across depths.

However, the correlation consistently declined with depth at most of the agriculture and mixed agriculture sites. The decline most often became more pronounced after the second (or sometimes third) layer indicating that the influence of the parame-

terized lower boundary extends through the lower 50 cm of the profile, at least to some extent. The use of a constant lower boundary condition would not be expected to correlate with variable observation.

### 4.3  Intercomparison of Noah, POME with *In-situ* Observations

The POME profiles have been compared with Noah LSM across the study region and against *in-situ* observations at ten locations. However, as mentioned earlier, both analyses have some limitations either in terms of proxy ground truth (in case of

LSM) and spatial representation (*in-situ* observations). Therefore, in this section an intercomparison between the three datasets is performed to assess the relative strength of each SM dataset. Figure 5 shows the time series of the SM state from Noah LSM, SCAN observations and the POME model. Consistent with the layer depths of the Noah, the POME profile and the SCAN observations were aggregated to 0-10; 10-40; and 40-100 cm layer depths.

The figure shows that the three data sets track each other well in some cases and that in others there are significant discrep-

ancies. In general, with the exception of a few cases, it appears that the POME and SCAN profiles track better than the Noah simulations in the lower soil layers. In particular, the Noah model shows a prominent cycling effect in the bottom layer at many sites that is not present in the other two data sets. On the other hand, the POME time series at the sites dominated by agricultural and mixed land uses (e.g., **2027, 2053, 2078, 2037, 2013**) exhibit a pronounced cyclic effect in the surface layers in addition to a positive bias compared to the Noah and the SCAN data. The sites in north Alabama (**2053** and **2078**) and in western Georgia

(**2027**) are located in some of the most highly irrigated regions of the study area. Therefore, the MW derived upper boundary condition for the POME model is likely to sense the irrigation activity in the region compared to the precipitation derived Noah LSM and SCAN observations. Further, the other two sites (**2037** and **2013**) are located in areas with close proximity to surface water and hence the MW surface SM estimates can be contaminated and result in the positive bias being observed at those locations. The forest dominated sites (2009, 2114 and 2115) on the other hand showed that the POME model profiles are more

comparable overall to the Noah LSM than to the point sourced SCAN observations. The mixed agricultural sites (**2113** and **2038**) are located in regions hardly experiencing any irrigation activity and not impacted by surface water, and thus showed little to no bias at the surface against Noah and SCAN profiles. The time series observations suggests that the downscaled MW surface SM is able to sense irrigation activity and therefore may provide an accurate surface boundary condition to the POME

model. At lower depths (40-100 cm) the POME profile seems to be in better agreement with the SCAN observations than the Noah estimates, which indicate that the parameterization of the lower boundary condition seems to be reasonable in this case, and that the POME profiles actually improve with depth in contrast to the Noah simulations. The dynamic nature is observed at lower depths for few sites (particularly **2013** and **2115**). This anomalous behaviour can be attributed to the fact that these sites had porous soils near the surface with clay layers beneath. While site **2113** is located in close proximity to a stream and is

potentially influenced by surface water intrusion.

Table-2 shows the detailed statistics of comparison between Noah LSM SM, *in-situ* observations and POME profiles at each SCAN site location. The results are further summarized across all sites in Fig. 6. The overall results show that the satellite-based and LSM SM estimates are reasonably comparable based on error statistics of ubRMSD (0.05 vs 0.04 $m^3m^{-3}$) and absolute bias (0.08 vs 0.07 $m^3m^{-3}$). For the surface layer (0-10 cm) comparisons, the Noah correlations are superior to the

POME model ($r = 0.75$ vs 0.54), although in several cases the Noah correlations decrease vertically through the soil column to the point that the two approaches are much more comparable (Fig. 6). This case does not show the steep decline in correlation through the POME profiles as before, indicating that amalgamation of the lower layers into one 60 cm layer has dampened that effect. In terms of mean bias across layers, the POME model is superior in four cases, Noah is superior in four cases and in the other two cases (2115 and **2053**) the two models perform the same. In terms of ubRMSD, the POME is superior to Noah

at three locations while at other six locations the difference is within 0.01 ($m^3m^{-3}$). Overall, the average statistics across all depths and all sites, the Noah average RMSD was 0.09 $m^3m^{-3}$ in comparison to the POME RMSD of 0.10 $m^3m^{-3}$ against ground based SCAN observations. The unbiased RMSD between Noah and SCAN was 0.04 $m^3m^{-3}$, and for the POME it was 0.05 in volumetric SM. Figure 6 shows that the Noah LSM tended to become less accurate compared to the SCAN observations with depth while the POME generally showed the reverse.

The three data sets can be further compared through TC analysis. TC has the advantage that the SCAN observations are treated equally with the LSM and POME as just another estimate of the true SM state. The analysis is performed for three layers to be consistent with the LSM model configuration (Fig. 7). The surface results (0-10 cm) showed that in most instances the SCAN observations are closer to the true SM compared to the Noah and POME data; however, the latter two data sets also show high coefficient of determination ($R^2$) values at several sites. The middle and bottom layer results appear to indicate that

the Noah LSM is superior (with 5 and 9 instances of $R^2 > 0.8$, respectively), while the SCAN observations and the POME model track each other fairly well with six and five instances, respectively, of $R^2 > 0.4$ for the POME and five and four such instances for SCAN observations. It should be noted here that although TC accounts for total error among the three data sets, the Noah results may be problematic in that, unlike the other two data sets, the deterministic SM equation (e.g., Richards Equation) governs the movement of moisture through the column and some of the random errors are eliminated. This would

not affect the surface layer, which is governed by precipitation and surface evaporation. Thus, the errors in the LSM at the deeper layers may be dampened and thus affect the results.

## 5   Error Characterization

The developed profile results are impacted by the boundary conditions applied to the POME as the integral serves to transition
the profile between the upper and lower boundary conditions. The upper boundary is associated with the MW surface SM estimates while the lower boundary was assumed for this study and potentially could be parameterized or used as a calibration parameter. In addition, the mean SM estimated from ALEXI determines the total mass to be distributed. Earlier studies by Al-Hamdan and Cruise (2010) and Mishra et al. (2015) showed that the POME model is capable of producing profiles with significant accuracy with mean absolute errors in the range of 0.5-3.0% for known input conditions. However, in this study
inputs to the POME model are derived from remotely sensed measurements, in addition to a parameterized bottom boundary condition. Hence, profile errors may be characterized in terms of errors in input parameters.

Figures 8(a) and (b) shows the sensitivity of the profile in terms of bias and RMSD to variations in the mean and surface constraints. From Figure 8(a) it is clear that, even if the surface boundary condition is off by 50% (in effective SM), the overall profile RMSD and bias is less than 0.35 (in effective SM), and the maximum possible deviation in the surface boundary results
in bias and RMSD of 0.62 and 0.67 (in effective SM), respectively. The sensitivity study of the mean moisture content (Fig. 8(b)) shows that the bias and RMSD of the profile (in terms of effective SM) are linearly related to the deviations in the assumed mean. Furthermore, Fig. 8 indicates that the profile is more sensitive to errors in the mean than it is to deviations in the surface boundary condition.

### 5.1   Effect of Disaggregation of AMSR-E MW Data

Figure 8 shows that the POME profile is sensitive to the surface boundary conditions. In this study these conditions are provided by AMSR-E; therefore, it is instructive to examine the relative accuracy of the downscaled MW data. To that end, the AMSR-E surface SM before and after disaggregation is compared to both the Noah LSM and the *in-situ* SCAN data to quantify the effect of the SEE downscaling algorithm. The results from a temporal analysis between coarse and downscaled (fine) resolution MW surface SM with the Noah LSM surface is shown in Fig. 9 for the study domain. The figure shows that the generally negative
bias of the original AMSR-E data (overall mean = -0.08 $m^3m^{-3}$) when compared to the Noah LSM was transformed by the disaggregation to a positive bias in the eastern half of the study area although the overall bias remained slightly negative. The positive bias in the eastern zone was largely in the 0.04 to 0.13 $m^3m^{-3}$ range. It is also apparent that this same area exhibited a substantial increase in correlation between the downscaled MW and Noah data. Comparing Fig. 9 to the land cover image in Fig. 1, it can be see that the increase in correlation was largely in the agricultural bands in the southwestern Georgia
leading into southeastern Alabama. However, a few areas, such as extreme southwestern and east-central parts of Alabama, showed degradation in correlation on downscaling. The land cover map shows that these areas are generally forested. Overall the temporal $r$ showed a modest increase from 0.21 to 0.25 with downscaling for the study area indicating that downscaled

AMSR-E is slightly more comparable to Noah LSM surface SM. Perusal of the figure shows that the poor results in Florida and along the eastern seaboard are primarily responsible for the low correlations. It also demonstrates the fundamental property that the downscaling process will be compromised in areas where the original MW data was of exceptionally poor quality to begin with.

5 It is difficult to determine the impact of the disaggregated MW surface SM estimates on the profiles compared to the LSM. First, the statistics shown in Fig. 9 are for the sensing depth of the raw AMSR-E data (0-2 cm) while the relatively better statistics shown in Fig. 2 are for the top layer corresponding to the Noah LSM (0-10 cm). This disparity in depth is undoubtedly affecting the results. The introduction of the mean SM from ALEXI also affects the near surface layer in the POME profile since mass balance must be maintained throughout the soil column. In any case, comparison of Fig. 2 and 9 shows that the

10 profile statistics are considerably improved compared to the MW surface values and thus the noise in the MW data has a minimal effect when compared to the Noah LSM.

 The results of the comparison with the SCAN sites are perhaps more instructive and are given in Table 3 below. The table shows that in terms of correlation, the disaggregated data were better related to the *in-situ* data than were the original coarse scale MW data ($r$ of 0.53 vs 0.31). This result was particularly evident at the agricultural SCAN sites ($r$ of 0.64 vs 0.42). These

15 results were obtained at a slight cost in the bias (bias: 0.07 vs -0.02 $m^3m^{-3}$) and RMSD (0.12 vs 0.10 $m^3m^{-3}$), although the difference was not as great in unbiased RMSD. In the case of Table 3, the SCAN depth is the same as the MW so comparisons are apt. In cases of relatively high bias in the MW data (e.g., sites 2009, 2114, **2053, 2078**) this error is introduced into the POME profile. Figure 8 shows that errors in the surface boundary of about 0.1 translate to bias and RMSD in the profile of about 0.05 (in effective SM). It appears from Table 3 that at the sites demonstrating the consistently higher bias and RMSD,

20 the error in the surface boundary could be responsible for one third to one half of that total.

## 5.2 Effect of Mean SM Inputs

The mean SM content within soil column in this study obtained from TIR based ALEXI model served as one of the two remotely sensed input parameters for the POME model. Therefore the mean SM content retrieved from the ALEXI model is compared with the Noah LSM. The results of the temporal analysis between the two datasets are shown in Fig. 10. The overall

25 bias between the two datasets is 0.04 $m^3m^{-3}$. The overall RMSD is 0.08 $m^3m^{-3}$ with ubRMSD of 0.04 $m^3m^{-3}$ indicating that the mean SM content of the two datasets is similar. In terms of correlation coefficient, the root zone correlation nearly doubled ($r = 0.49$) compared to the surface correlations (Fig. 9). Further, comparison of Fig. 10 with Fig. 1 reveals that, similar to the surface SM analysis, the mean SM content with the highest correlations ($r > 0.5$) are observed mostly in agriculture-dominated areas.

30 Figure 8(b) shows that the translation of the error in the mean SM content to errors in the POME profile is linear, so an error of 0.04 in the ALEXI mean compared to the LSM would translate into a similar error in the computed profile. Examination of column 2 (NP) in Table 2 above shows that this error represents the majority of the errors in the computed POME profiles compared to the LSM.

Overall, the analysis revealed that the surface SM estimates accounted for, at most, one third to half of the error in the SM profiles developed using the POME model. For most of the cases, the mean SM content and the parameterized lower boundary accounted for the majority of the error. Recent advances such as the L-band sensor aboard the SMAP mission, offers the potential for even better correlated MW data. In addition, further analysis of the lower boundary condition parameterization could improve the profiles, particularly in the lower layers. As discussed earlier, since the ultimate purpose of the developed profiles is to be assimilated into a LSM, the lower boundary can serve as a highly effective way to tie the POME profile to the model climatology by using the model SM in the lowest layer of the soil column as the lower boundary on the POME integral. Since this level ( 100-200 cm) is normally well below the root mass of most vegetation species (particularly row crops) (Wu et al., 1999), then its selection will have minimal impact on the LSM model results. For instance, Mishra et al. (2013) used POME generated profiles to update SM within a crop model using the lower boundary condition from the crop model itself. If sufficient ground truth data are available, calibration could be accomplished, or the lower boundary could be set as a function of soil properties in the bottom layer of the profile.

The relatively sparse (5-10 day recurrence interval) availability of the ALEXI TIR-based SM retrieval is the major weakness of the procedure and necessitated compositing of the data into three day running means. However, the issue is a function of the semi-tropical humid climate of the Southeastern U.S. Drier regions of the world would not suffer as much from this issue. Thus it is possible that the proposed method could be employed to deduce vertical SM profiles in regions of the world where observed climate data are scarce or insufficient to drive ecological models. These profiles could be assimilated into such models to help correct for bias due to the poor climate inputs.

## 6 Conclusions

This study evaluated the feasibility of linking downscaled MW surface SM with TIR root zone estimates to develop entropy-based vertical SM profiles. The SM profiles (including surface values) were compared to *in-situ* data at the Southeastern U.S. as well as the Noah LSM within the NASA LIS. Initial results are encouraging. The SEE disaggregation method of Merlin et al. (2012), guided by high resolution TIR estimates from the ALEXI model, showed promise when compared to the *in-situ* and modeled estimates in a humid semi-tropical region of the U.S. The POME generated SM profiles generally compared favorably with the SCAN site profiles and the Noah LSM.

When the Noah LSM and the POME profiles were compared to the *in-situ* data in terms of bias, the POME-generated profiles were clearly superior at four sites, the LSM was superior at four sites and the two methods were similar at the other sites. However it must be noted that there is a scale mismatch when comparing with *in-situ* observations, also biases may exist in SCAN observations as well as the models. The maximum correlation in the range of 0.4-0.65 was observed in agriculturally dominant areas. Further the highest correlations were found at the depth of 10-40 cm, coinciding with the maximum root density for crops and thus offering a better coupling between land and atmosphere. The ALEXI model was able to pick the wetting and drying trends in the root-zone consistently.

Compared to *in-situ* observations, the bias and RMSD of the Noah model often tended to degrade vertically with depth while the reverse was evident in most of the POME profiles. While acknowledging that the SCAN data contains biases and errors of their own, this characteristic of the remote sensing driven POME method seems to open the possibility that profiles from LSMs could be improved in terms of bias and RMSD through the assimilation of the remotely sensed profiles. This conclusion is bolstered by the results of previous studies [e.g., Al-Hamdan and Cruise (2010); Singh (2010); Mishra et al. (2015)] have found that the when the input data are well defined, the POME profiles are highly accurate throughout the soil column. TC analysis revealed that the POME and observed SCAN site observations tracked well, while the LSM appeared to show less variability, possibly due to the use of the deterministic Richards Equation to model SM movement through the soil column.

Error analyses revealed that the majority of the error in the POME generated profiles was due to error in the mean SM deduced from the ALEXI retrievals and the parameterized lower boundary condition. The lower boundary was simply assumed to be 50% of available moisture capacity and this assumption proved to be incorrect in some cases where the lower layer in the soil column exhibited marked temporal variability. The SEE downscaling procedure increased the correlation of the surface SM compared to both the LSM and the SCAN sites, especially in agricultural areas where correlations in the range of 0.5-0.8 were achieved. In the meantime, the overall bias was reduced by a factor of 4 and the RMSD was only slightly increased (0.09 to 0.10 $m^3m^{-3}$). The error analysis suggested that at most the MW derived surface boundary accounted for at most one third to one half of the overall error in the POME profiles. Downscaling generally was less effective in locations where the AMSR-E demonstrated positive bias and appeared to lose effectiveness as the bias increased. MW surface observations can be contaminated when a high percentage of the pixel is dominated by water, as near large streams or lakes or in the near coastal region. Dense vegetation also tends to degrade the MW results.

## Appendix A: The Principle of Maximum Entropy (POME) Model

Three distinct SM profiles cases are possible: wet case- upper layer is wetter than the lower; dry case – upper layer is drier than the lower; and dynamic case: a mix of wet and dry case, typically has at least one prominent inflection point. The wet case arises right after an irrigation/precipitation event; dry case profiles result long time after precipitation/irrigation event. The dynamic case is caused some time after a precipitation/irrigation event when surface dries up but the middle layers still have elevated moisture contents from earlier precipitation/irrigation events. Figure A1 shows these three possible cases.

### A1   Application of Entropy to SM Profile Development

The information in a system in state $i$ according to classical Shannon entropy formulation is given as:

$$I_i = ln\left(\frac{1}{p_i}\right) \tag{A1}$$

Here $p_i$ is the probability that the system is in state $i$, and $ln$ is the natural logarithm. The mean information content of a variable or process is given by Shannon entropy formulation in continuous form:

$$H = - \int_{i=1}^{n} f(x) ln\big(f(x)\big) dx \tag{A2}$$

Here, $f(x)$ is continuous probability distribution function (pdf) and $n$ is the number of values a variable can take. Higher values of entropy would mean the lesser prior information availability in other words higher entropy means the higher uncertainty. According to Mays et al. (2002), the informational entropy is a measure of the correspondence between the pdf of the dataset associated with a system and the pdf associated with the minimum information about the system. In case, the minimum *a-priori* information is available or the system is unpredictable, the probability distribution would be uniform and the entropy will be high (Pachepsky et al., 2006).

The principle of maximum entropy was developed by Jaynes (1957a, b) which proposes that if inferences has to be drawn from incomplete information then it should be drawn from a probability distribution that has the maximum entropy permitted by the a-priori information. Based on this concept, Al-Hamdan and Cruise (2010) developed an entropy-based algorithm to model SM profiles assuming uniform distribution throughout the soil column. The method was based on the approach developed to compute the vertical velocity distribution in open channels by Chiu (1987).

The application of POME to develop a one-dimensional SM profile requires two constraints; total probability:

$$\int_{\Theta_L}^{\Theta_0} f(\Theta) d\Theta = 1 \tag{A3}$$

and the mass balance constraint:

$$\int_{\Theta_L}^{\Theta_0} \Theta f(\Theta) d\Theta = \overline{\Theta} \tag{A4}$$

Here $\Theta$ is the effective saturation and $\overline{\Theta}$ is the mean moisture of the soil column; whereas $\Theta_0$ and $\Theta_L$ are the upper (surface) and lower (bottom) effective saturation. The effective SM is given as: $(\theta - \theta_{wp})/(\theta_{fc} - \theta_{wp})$. The second constraint serves to connect the first moment in probability space to the mean water content of the soil column in physical space. The Shannon entropy is given by Shannon (1948):

$$I = - \int_{0}^{\infty} f(x) ln\big(f(x)\big) dx \tag{A5}$$

Maximizing $I$ in Eq. (A5) for the uniform pdf subject to the constraints, Chiu (1987) developed the 1-D profile of a variable decreasing monotonically from the surface down using the method of Lagrange multipliers. Al-Hamdan and Cruise (2010) applied the same technique to develop vertical SM profiles either increasing or decreasing with depth from the surface:

$$\Theta(z) = \frac{ln[exp(\lambda_2 \Theta_0) \pm exp(1 - \lambda_1)(\frac{z}{L})]}{\lambda_2} \tag{A6}$$

The Lagrange multipliers ($\lambda's$) can be determined from application of the constraints and boundary conditions (surface effective saturation, $\Theta_0$) and mean effective saturation value of the soil column $\overline{\Theta}$, $z$ is calculation depth, and $L$ is total depth of the column. Equation (A6) is a monotonically increasing (+ sign) or decreasing (- sign) function, representing dry (increasing from the top boundary) and wet (increasing from the bottom boundary) case profiles (Fig. A1). The dynamic case can be handled by dividing the profiles into multiple monotonic profiles and running the wet/dry case models for each of those profiles separately while keeping the mass balance.

**Appendix B: Soil Water Characteristics**

The soil water characteristics used in this study to map effective to volumeteric SM and *vice-versa* is obtained from the values reported by Rawls et al. (1982) in their study. Table A1 summarizes the properties used for each soil types.

*Acknowledgements.* The authors would like to express their sincere gratitude to the reviewers for their constructive comments towards the improvement of this manuscript. This study is supported by NASA Headquarters under the NASA Earth and Space Science Fellowship (NESSF) Program - Grant NNX15AN58H. The authors are also thankful of agencies such as NASA, NRCS for making soil moisture and other data publicly available.

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

## Tables

**Table 1.** SCAN site 5 km dominant land cover (MODIS 2008) and soil characteristics (SCAN) at surface and depth of 100 cm [S-Sand; L-Loam; C-Clay and Si-Silt]. Soil information was not available for sites 2037 and 2038.

| SCAN Site | Lat/Lon | Land cover | Soil Type (SCAN) Surface | 100cm |
|---|---|---|---|---|
| 2009 | 30.3/-84.4 | Savannas/Mix Forest | S | S |
| 2013 | 33.8/-83.4 | Crop/Savannas | SL | C |
| 2027 | 31.5/-83.5 | Cropland | S | SL |
| 2037 | 34.3/-79.7 | Crop/Shurbland | - | - |
| 2038 | 32.6/-81.2 | Crop / Savannas | - | - |
| 2053 | 34.9/-86.5 | Cropland | SiCL | SiC |
| 2078 | 34.9/-86.6 | Cropland | SiCL | C |
| 2113 | 34.2/-86.8 | Crop/Savannas | L | SCL |
| 2114 | 32.6/-88.2 | Savannas | SCL | CL |
| 2115 | 32.4/-85.7 | Savannas | LS | SC |

**Table 2.** Results of temporal comparisons in absolute bias($m^3 m^{-3}$), RMSD ($m^3 m^{-3}$), ubRMSD ($m^3 m^{-3}$) and correlation at 10 sites between the developed profile and Noah SM profiles against SCAN observations at 0-10; 10-40 and 40-100cm depths [NP - Noah vs POME; SP - SCAN vs POME; and NS – Noah vs SCAN]

| Site | Bias | | | RMSD | | | ubRMSD | | | Correlation | | | |
|------|------|------|------|------|------|------|------|------|------|------|------|------|---|
| | NP | SP | NS | NP | SP | NS | NP | SP | NS | NP | SP | NS | |
| 2009 | 0.02 | 0.13 | 0.11 | 0.05 | 0.14 | 0.12 | 0.04 | 0.04 | 0.03 | 0.12 | 0.23 | 0.56 | |
| 2014 | 0.00 | -0.10 | -0.10 | 0.06 | 0.13 | 0.11 | 0.06 | 0.07 | 0.03 | 0.54 | 0.50 | 0.85 | |
| 2015 | 0.01 | 0.15 | 0.14 | 0.07 | 0.16 | 0.14 | 0.07 | 0.06 | 0.00 | 0.50 | 0.54 | 0.72 | |
| 2027 | 0.09 | 0.19 | 0.10 | 0.10 | 0.20 | 0.11 | 0.06 | 0.07 | 0.05 | 0.64 | 0.49 | 0.70 | |
| 2053 | 0.05 | 0.03 | -0.02 | 0.07 | 0.06 | 0.04 | 0.05 | 0.06 | 0.03 | 0.77 | 0.75 | 0.85 | 0-10 cm |
| 2078 | 0.05 | 0.03 | -0.02 | 0.09 | 0.09 | 0.05 | 0.07 | 0.08 | 0.05 | 0.73 | 0.69 | 0.72 | |
| 2113 | 0.00 | 0.02 | 0.03 | 0.06 | 0.08 | 0.06 | 0.06 | 0.08 | 0.05 | 0.41 | 0.51 | 0.86 | |
| 2037 | 0.06 | 0.08 | 0.02 | 0.09 | 0.10 | 0.04 | 0.07 | 0.06 | 0.03 | 0.40 | 0.63 | 0.72 | |
| 2038 | 0.02 | 0.02 | 0.16 | 0.06 | 0.06 | 0.04 | 0.05 | 0.05 | 0.04 | 0.34 | 0.37 | 0.62 | |
| 2013 | 0.04 | 0.07 | 0.23 | 0.07 | 0.09 | 0.04 | 0.05 | 0.05 | 0.03 | 0.59 | 0.67 | 0.88 | |
| 2009 | 0.06 | 0.18 | 0.12 | 0.08 | 0.19 | 0.13 | 0.05 | 0.03 | 0.05 | 0.21 | 0.17 | 0.37 | |
| 2014 | 0.02 | -0.14 | -0.16 | 0.05 | 0.14 | 0.17 | 0.04 | 0.04 | 0.06 | 0.69 | 0.60 | 0.78 | |
| 2015 | 0.00 | 0.04 | 0.04 | 0.04 | 0.06 | 0.05 | 0.04 | 0.04 | 0.03 | 0.52 | 0.51 | 0.80 | |
| 2027 | 0.06 | 0.14 | 0.08 | 0.07 | 0.14 | 0.09 | 0.04 | 0.05 | 0.04 | 0.70 | 0.56 | 0.63 | |
| 2053 | 0.00 | 0.03 | 0.03 | 0.05 | 0.07 | 0.06 | 0.05 | 0.06 | 0.05 | 0.67 | 0.51 | 0.74 | 10-40 cm |
| 2078 | 0.01 | -0.06 | -0.06 | 0.05 | 0.07 | 0.09 | 0.05 | 0.05 | 0.06 | 0.69 | 0.56 | 0.56 | |
| 2113 | 0.04 | 0.03 | -0.01 | 0.08 | 0.06 | 0.03 | 0.06 | 0.05 | 0.03 | 0.37 | 0.37 | 0.91 | |
| 2037 | 0.07 | 0.10 | 0.02 | 0.08 | 0.10 | 0.03 | 0.03 | 0.03 | 0.02 | 0.57 | 0.55 | 0.78 | |
| 2038 | 0.08 | -0.01 | -0.09 | 0.09 | 0.04 | 0.10 | 0.05 | 0.04 | 0.04 | 0.44 | 0.39 | 0.55 | |
| 2013 | 0.07 | 0.05 | -0.02 | 0.10 | 0.08 | 0.04 | 0.07 | 0.06 | 0.03 | 0.29 | 0.27 | 0.88 | |
| 2009 | 0.09 | 0.20 | 0.11 | 0.10 | 0.20 | 0.12 | 0.05 | 0.03 | 0.05 | 0.22 | 0.16 | 0.33 | |
| 2014 | 0.04 | -0.17 | -0.21 | 0.08 | 0.17 | 0.22 | 0.07 | 0.02 | 0.07 | 0.65 | 0.59 | 0.68 | |
| 2015 | 0.00 | -0.02 | -0.02 | 0.04 | 0.08 | 0.06 | 0.04 | 0.07 | 0.05 | 0.48 | 0.45 | 0.81 | |
| 2027 | 0.08 | 0.05 | -0.02 | 0.10 | 0.06 | 0.06 | 0.06 | 0.03 | 0.06 | 0.38 | 0.40 | 0.58 | |
| 2053 | 0.01 | -0.04 | -0.04 | 0.08 | 0.06 | 0.06 | 0.08 | 0.05 | 0.05 | 0.49 | 0.51 | 0.85 | 40-100 cm |
| 2078 | 0.03 | -0.08 | -0.11 | 0.08 | 0.09 | 0.13 | 0.08 | 0.04 | 0.06 | 0.54 | 0.54 | 0.74 | |
| 2113 | 0.07 | -0.03 | -0.10 | 0.11 | 0.06 | 0.11 | 0.08 | 0.05 | 0.05 | 0.32 | 0.27 | 0.85 | |
| 2037 | 0.09 | 0.04 | -0.04 | 0.10 | 0.05 | 0.06 | 0.04 | 0.03 | 0.03 | 0.52 | 0.45 | 0.68 | |
| 2038 | 0.13 | -0.10 | -0.22 | 0.14 | 0.10 | 0.23 | 0.06 | 0.03 | 0.06 | 0.42 | 0.40 | 0.35 | |
| 2013 | 0.12 | 0.08 | -0.04 | 0.15 | 0.12 | 0.09 | 0.09 | 0.09 | 0.07 | 0.16 | 0.10 | 0.65 | |

**Table 3.** Statistical comparison before and after disaggregation of coarse resolution MW SM against SCAN observations [r – correlation coefficient;bias, RMSD and ubRMSD in $m^3m^{-3}$; N – number of days data points was available; maximum possible N = 1825];*non-significant correlation using two-tailed t-test at 99% CI*

| Site | Mean SM | | | | SCAN/MW(25km) | | | | SCAN/MW(5k) | | | |
|------|------|-----------|----------|------|------|------|--------|--------|------|------|--------|------|
| | SCAN | MW (25k) | MW (5k) | N | Bias | RMSD | ubRMSD | r | Bias | RMSD | ubRMSD | r |
| 2009 | 0.06 | 0.18 | 0.19 | 841 | 0.12 | 0.12 | 0.02 | -0.12* | 0.13 | 0.14 | 0.04 | 0.17 |
| 2014 | 0.26 | 0.15 | 0.19 | 1055 | -0.11 | 0.14 | 0.09 | 0.30 | -0.07 | 0.12 | 0.09 | 0.47 |
| 2015 | 0.08 | 0.15 | 0.25 | 1103 | 0.08 | 0.09 | 0.04 | 0.42 | 0.17 | 0.19 | 0.08 | 0.60 |
| | **Mean** | | | | **0.03** | **0.12** | **0.05** | **0.22** | **0.08** | **0.15** | **0.07** | **0.42** |
| 2027 | 0.08 | 0.13 | 0.29 | 1241 | 0.05 | 0.06 | 0.03 | 0.48 | 0.21 | 0.22 | 0.08 | 0.50 |
| 2053 | 0.24 | 0.14 | 0.32 | 1160 | -0.10 | 0.13 | 0.08 | 0.43 | 0.08 | 0.10 | 0.06 | 0.74 |
| 2078 | 0.25 | 0.14 | 0.31 | 1080 | -0.12 | 0.13 | 0.05 | 0.34 | 0.06 | 0.10 | 0.08 | 0.68 |
| | **Mean** | | | | **-0.06** | **0.10** | **0.05** | **0.42** | **0.12** | **0.14** | **0.07** | **0.64** |
| 2113 | 0.20 | 0.14 | 0.18 | 1014 | -0.06 | 0.12 | 0.10 | 0.44 | -0.02 | 0.10 | 0.09 | 0.47 |
| 2037 | 0.16 | 0.14 | 0.21 | 1157 | -0.02 | 0.06 | 0.05 | 0.11 | 0.06 | 0.10 | 0.09 | 0.62 |
| 2038 | 0.14 | 0.16 | 0.16 | 1067 | 0.02 | 0.05 | 0.04 | 0.45 | 0.02 | 0.06 | 0.06 | 0.31 |
| 2013 | 0.21 | 0.15 | 0.23 | 1218 | -0.06 | 0.09 | 0.06 | 0.21 | 0.02 | 0.05 | 0.05 | 0.55 |
| | **Mean** | | | | **-0.03** | **0.08** | **0.06** | **0.26** | **0.02** | **0.08** | **0.07** | **0.53** |
| | **Overall Mean** | | | | **-0.02** | **0.10** | **0.06** | **0.31** | **0.07** | **0.12** | **0.07** | **0.53** |

**Table A1.** The soil water characteristic values used in this study based on the earlier study by Rawls et al. (1982)

| Soil Type | Residual $(m^3m^{-3})$ | Wilting Point $(m^3m^{-3})$ | Field Capacity $(m^3m^{-3})$ | Porosity $(m^3m^{-3})$ |
|---|---|---|---|---|
| Sand | 0.020 | 0.033 | 0.091 | 0.437 |
| Loamy Sand | 0.035 | 0.055 | 0.125 | 0.437 |
| Sandy Loam | 0.041 | 0.095 | 0.207 | 0.453 |
| Silt Loam | 0.015 | 0.133 | 0.330 | 0.501 |
| Silt | 0.020 | 0.110 | 0.370 | 0.481 |
| Loam | 0.027 | 0.117 | 0.270 | 0.463 |
| Sandy Clay Loam | 0.070 | 0.148 | 0.255 | 0.398 |
| Silty Clay Loam | 0.040 | 0.208 | 0.366 | 0.471 |
| Clay Loam | 0.075 | 0.197 | 0.318 | 0.464 |
| Sandy Clay | 0.109 | 0.239 | 0.339 | 0.430 |
| Silty Clay | 0.056 | 0.250 | 0.387 | 0.479 |
| Clay | 0.090 | 0.272 | 0.396 | 0.475 |

**Figures**

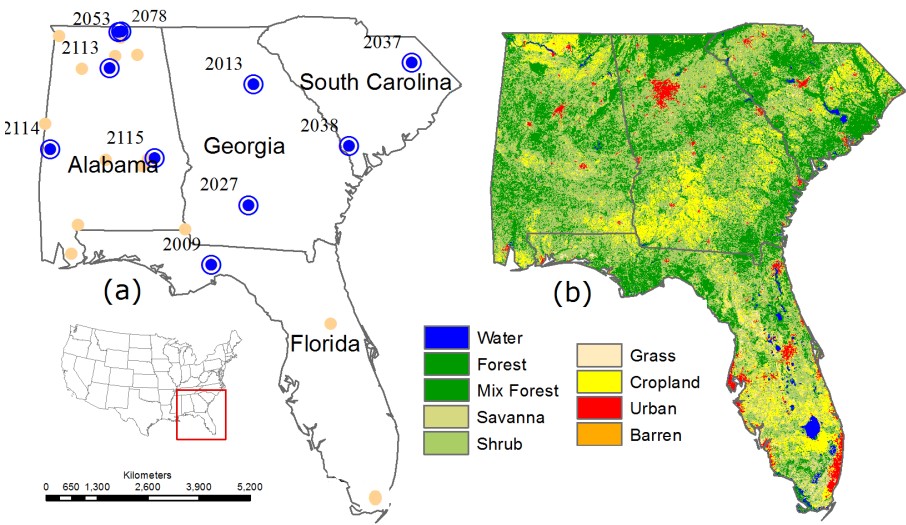

**Figure 1.** (a)Overview of study area showing location of all active SCAN sites. The dark blue circles indicate sites with most consistent data availability and are being used for comparison and validation in this study.(b) The figure shows a land cover map (MODIS-2008) for the study area.

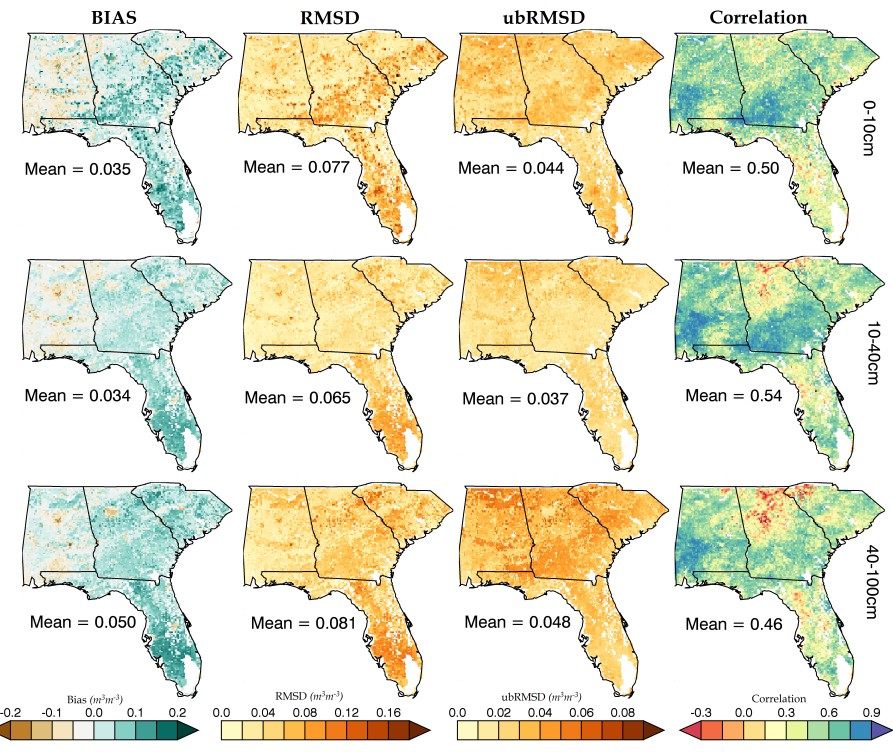

**Figure 2.** Map of bias, RMSD, ubRMSD and correlation coefficient between Noah and POME SM profiles over multiple years (2006-2010) at different layer depths: top panel: 0-10 cm; middle panel: 10-40 cm and bottom panel: 40-100 cm.

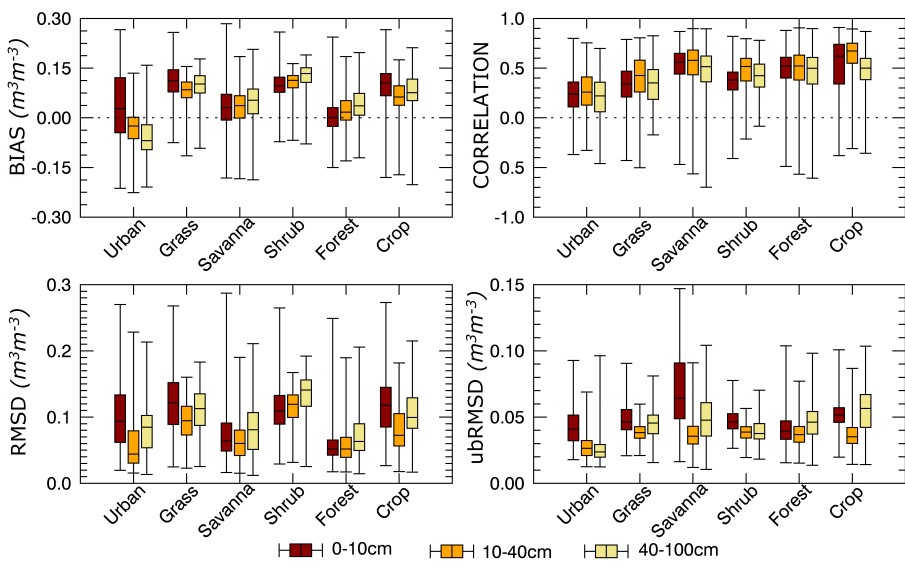

**Figure 3.** Comparison of Noah and POME SM profiles at multiple layer depths by Land Cover across Southeast United States

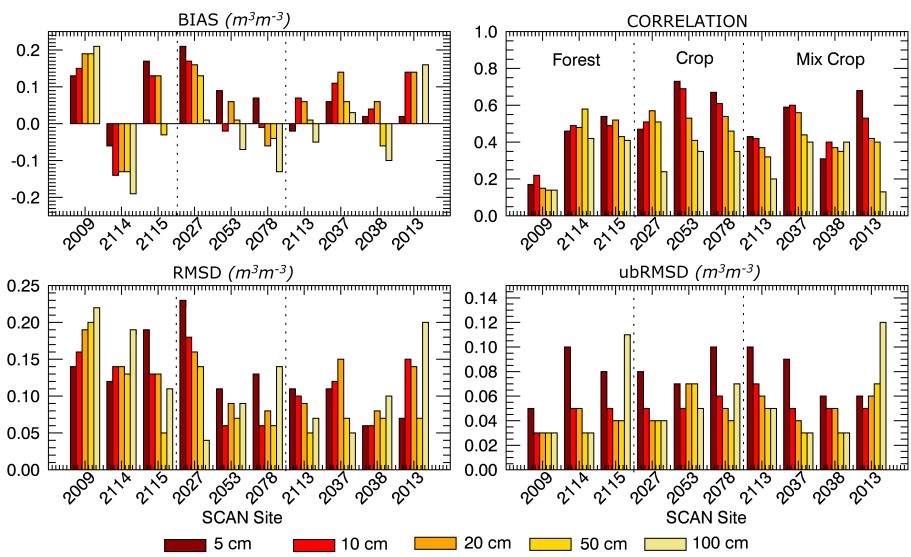

**Figure 4.** Statistics at SCAN sites showing bias, Correlation, RMSD and ubRMSD between SCAN observations and POME SM profiles at multiple layer depths.

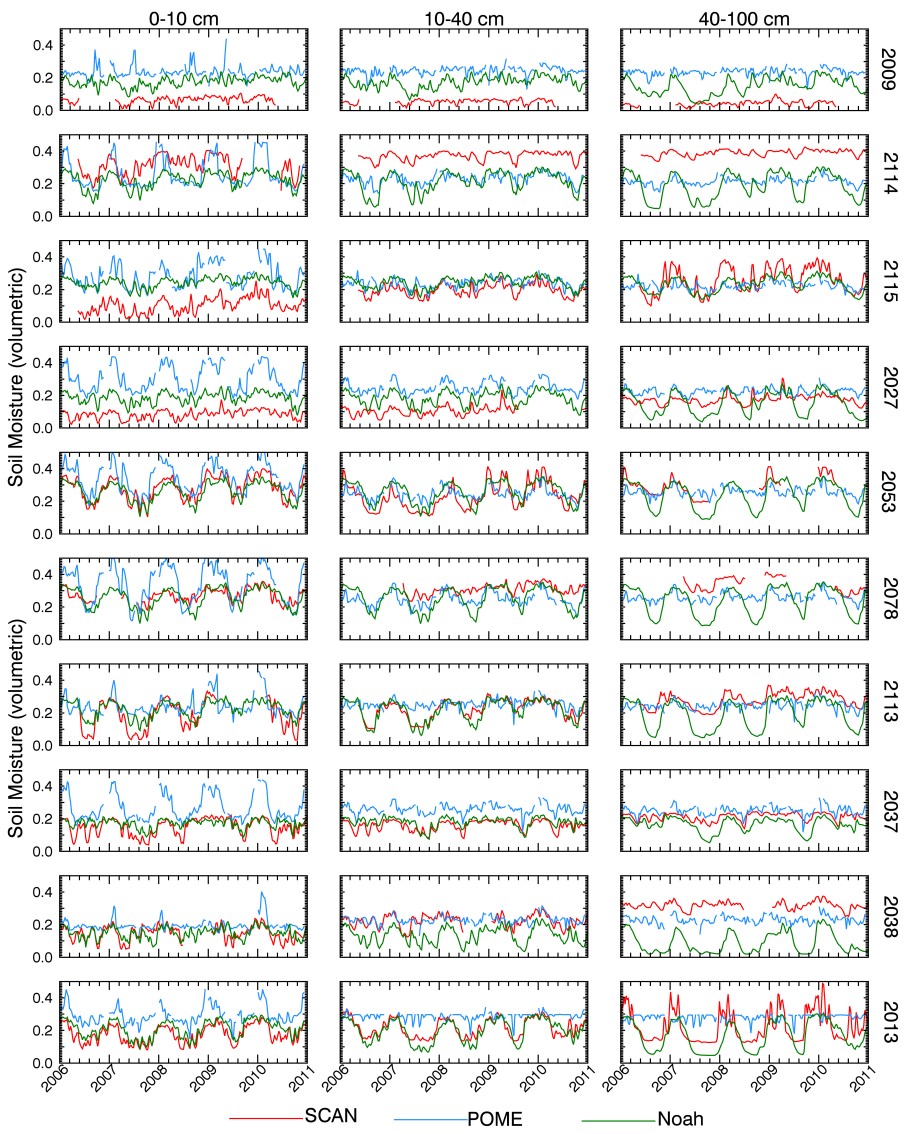

**Figure 5.** Time series of soil moisture condition (every 8-day interval) at 10 NRCS SCAN sites from the POME model (Blue); Noah LSM (green) and *in-situ* observations (red) at three layer depths (2006-2010)

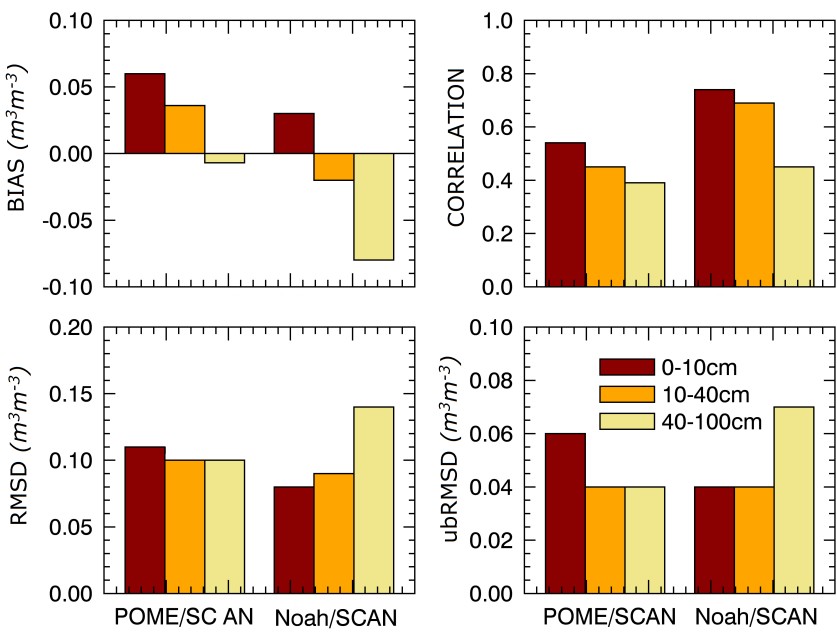

**Figure 6.** POME and Noah SM profiles statistics at all SCAN sites compared against *in-situ* observations averaged across layer depths

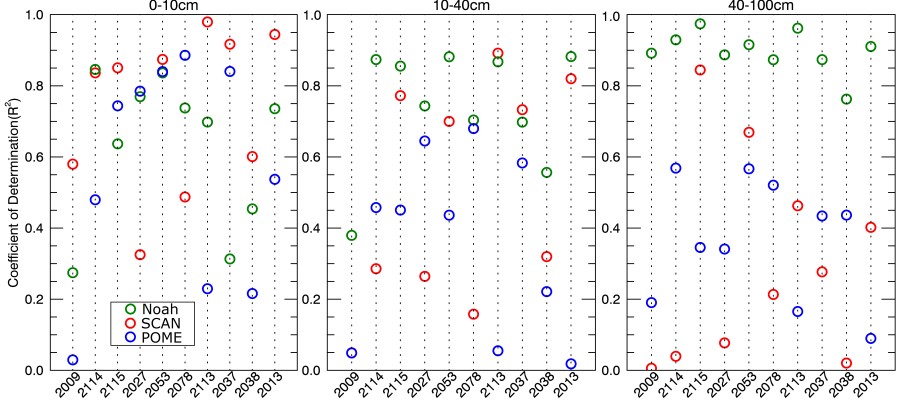

**Figure 7.** Triple collocation analyses of SM profiles from Noah (green), POME (blue) and *in-situ* observations (red) at scan site locations at the depths of 0-10; 10-40 and 40-100 cm

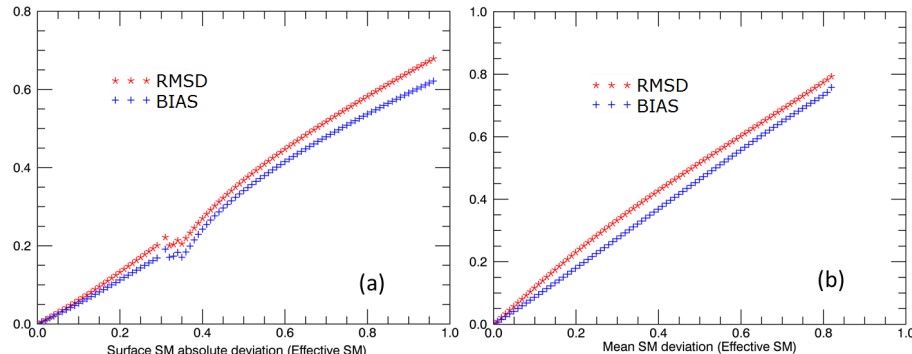

**Figure 8.** POME model sensitivity to (a) boundary condition; (b) sensitivity to profile mean input towards profile Bias and RMSD in terms of effective SM.

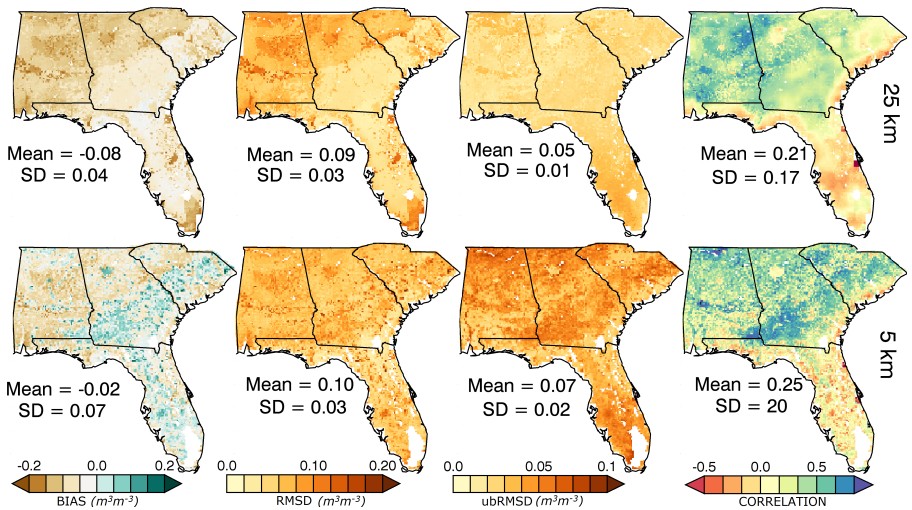

**Figure 9.** Map of Southeast United States demonstrating temporal statics in bias, RMSD, ubRMSD and correlation between coarse (top panel) and fine (bottom panel) resolution AMSR-E (MW) and Noah LSM surface SM (2006-2010)

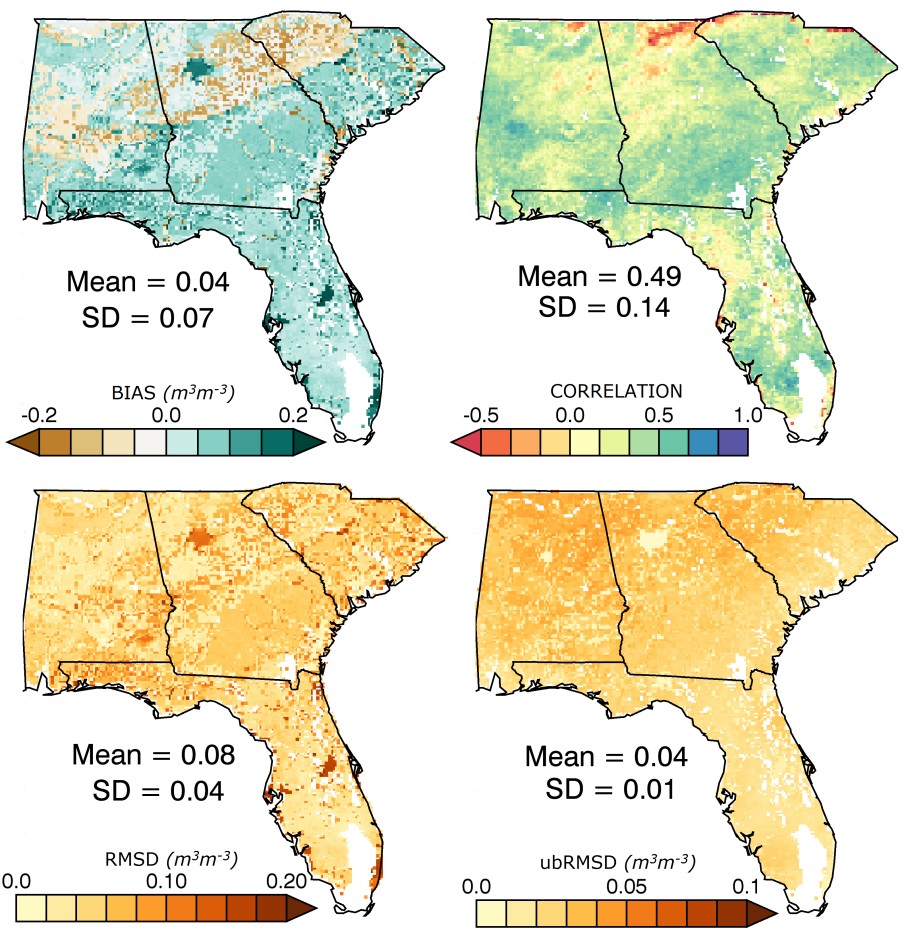

**Figure 10.** Map of temporal statistics between root zone ALEXI and Noah SM (2006-2010).

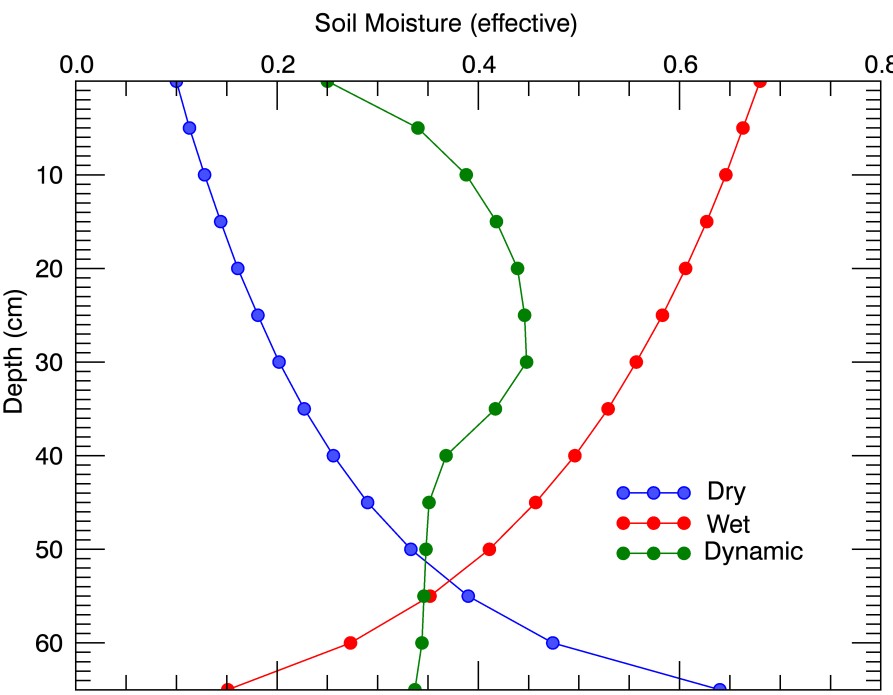

**Figure A1.** Plot displaying the three possible cases of the soil moisture profile