# Peer review of "Development of Soil Moisture Profiles Through Coupled Microwave-Thermal Infrared Observations in the Southeastern United States"

_Hydrology and Earth System Sciences, 2017_

## Author Comment (AC1) · 7 Sep 2017

We regret that there was an error in Tables-2 and 3 of our manuscript. In table-2 data from layer depth 10-40 cm is erroneously copied over to layer depth 40-100 cm. Then in table-3, site 2015 was erroneously identified as site 2013 in the third line of the table.

We have corrected the errors and uploaded the corrected tables as a supplement to this comment.

[Figure]

Please also note the supplement to this comment:
https://www.hydrol-earth-syst-sci-discuss.net/hess-2017-351/hess-2017-351-AC1-supplement.pdf

———————————————————————

---

## Referee Comment (RC1) · Anonymous Referee #1 · 23 Oct 2017

Here I reviewed "Development of Soil Moisture Profiles Through Coupled Microwave-Thermal Infrared Observations in the Southeastern United States" by Vikalp Mishra et al. The manuscript aims to develop a soil moisture profile estimation methodology based on remote sensing data. It may have some potential contributions to global scale root zone soil moisture estimation. However, some key information was missing in the figures, which made the manuscript hard to follow and evaluate. Further, the structure of the manuscript should be better organized.

General comments: 1. AMSR-E downscaling: I would first suggest the authors to

clarify their research goals. If providing high resolution data is not part of the goal, the authors can perform analysis at 25-km, i.e. upscale ALEXI. Downscaling MS is usually challenging. The consequences on POME are also difficult to evaluate, as pointed by the authors (page 16, line 535).

2. The proposed method can only handle cases with soil moisture is linearly increasing/decreasing with depth, if I am correct. If that is the case, the authors should discuss why the proposed method is preferable than other remote sensing based method, e.g. exponential filter (Albergel et al., 2008).

3. Please add units to all the figures

4. The conclusions should be presented in a more concisely.

Detailed comments: 1. Line 16 to 18: please add units to all the numbers being reported. I assume it is in m3/m3?

2. Line 67 to 69: please revise/modify the goal here. The authors should at least mention the methodology should satisfy what applications.

3. Section 2.1: Please specify why this area is selected. It is known that AMSR-E has the poorest performances over dense vegetation areas. This means AMSR-E is usually more accurate over southwest part of the CONUS and less accurate over the eastern part of the CONUS.

4. Section 2.2.1: Please justify why LPRM based C-band AMSR-E data were not used, since it is usually considered to have a better quality?

5. Equation 6 and 7: the author can remove one of them

6. The captions of figure 2 should be modified. Please specify which products are compared in the caption.

7. Line 343 and elsewhere: RMSE is the root mean square error. It can be calculated when you have a known "truth" or a very good reference. If NOAH is not assumed to

be perfect, I would suggest the author to change it into RMSD, i.e. root mean square difference.

8. Line 349: provide the unit here is m3/m3, I would not say ubRMSE = 0.06 m3/m3 is small. . .

9. Line 353: the author may need to define a threshold of "well" or "good". As shown in the third row of Figure 2, large fractions of correlations are below 0.4. It is hardly to be considered as "well" in my background. However, I agree that this threshold varies according to different applications.

10. Figure 5: please add row indices.

11. Line 476 to 477: please rephrase.

12. Line 478: the implementation of TC should include more information. Was the climatology removed from each dataset?

13. Line 484 and figure 7: it is not common to express TC results in R2. Please specify how this metric was derived.

14. Line 488 to 490 is incorrect. TC estimates the total error, instead of just the random error. Please see Yilmaz and Crow 2014. This means if NLDAS is less accurate, either due to random or temporally correlated errors, it will be shown in the TC results.

15. Section 4.4.1: please refer to my general comment. If the reviewer can perform analysis at coarse scales, this section is unnecessary. This may make the manuscript cleaner.

16. Line 578 to 579: I would not consider the minimum bias is the key advantage of POME. This is because it is nearly impossible to define an absolute bias at large scales, since the reference dataset (e.g. SCAN) can also be biased.

17. Line 585: Root zone soil moisture at large scales can have significant spatial variability, according to my experience. It can result in large errors/bias using limited

point sensors to represent large scale root zone soil moisture. Hence, I'm suspecting how confidently the authors can draw this conclusion.

18. Line 598 to 600: please refer to my General Comment 1

References: Albergel, Clément, et al. "From near-surface to root-zone soil moisture using an exponential filter: an assessment of the method based on in-situ observations and model simulations." Hydrology and Earth System Sciences Discussions 12 (2008): 1323-1337. Yilmaz, M. Tugrul, and Wade T. Crow. "Evaluation of assumptions in soil moisture triple collocation analysis." Journal of Hydrometeorology 15.3 (2014): 1293-1302.
* * *

---

## Referee Comment (RC2) · Anonymous Referee #2 · 1 Nov 2017

This study presents the application of the principle of maximum entropy model (POME) to estimate soil moisture profiles from remote sensing data for the area of the southeastern United States. The ultimate goal of the approach is to estimate soil moisture profiles from remote sensing data exclusively without the need of onsite measurements or additional information obtained from a land surface model. The only input data required by POME are ground surface soil moisture content (in this case derived from downscaled AMSR-E) data and profile mean soil moisture content estimated from thermal-infrared data using the ALEXI surface energy balance model. POME-

estimated soil moisture profiles are compared to soil moisture profiles simulated with the Noah land surface model and in situ data from 10 sites (SCAN sites) distributed throughout the study area and representing different land uses. Comparison is done by simple statistics (bias, RMSE, unbiased RMSE and correlation coefficient), whereas mean values and, in some cases, pixelwise values are evaluated and advantages and limitations of the different models and datasets are critically discussed.

Overall the paper is very well written and the text is well supported by appropriate, highquality figures. I am a soil hydrologist by education and I am not or only little familiar with most of the models and datasets applied in this study, so it is rather challenging to judge whether the application of the presented models and data products is a good choice or reasonable. In this respect, I have to rely on the experience of the authors. Nevertheless, I have some comments I would like to have elaborated or explained in a revised version of the manuscript:

POME model: I would still like to have some more information about the model. It only requires surface soil moisture and soil profile mean soil moisture as input information and assumes that moisture is either monotonically increasing or decreasing. If this is not the case, an inflection point that is located in the soil layer with the highest field capacity is assumed

- How does the POME model consider several layers of different texture? They might exhibit rather different soil moisture contents with distinct "jumps" between layers. In addition, infiltration fronts might induce further "inflection points". I cannot really imagine how this can be estimated from surface soil moisture and mean profile water content only. Will there be just one optimal soil moisture profile or might there be several (concept of equifinality).

- How is the lower boundary condition parameterized?

As the authors state correctly, both the Noah and the POME model will be affected by model errors and consequently may provide erroneous soil moisture profiles while
the in situ data were measured at a much smaller measurement scale and the measurement site may not be representative for the entire pixel. Nevertheless, for a soil hydrologist, the time series shown in Figure 6 provide the most valuable information for identifying model errors and weaknesses of models and data. Unfortunately, this figure is almost not discussed within the manuscript. Observations to be addressed are e.g.

- Except some sites where the SCAN data show some bias (mostly towards the dry range), SCAN and Noah data appear to agree rather well, both in absolute soil moisture content and in the amplitude of the annual dynamics whereas POME often shows a very strong dynamics (especially towards the wet range). Might this cause problems with the upper boundary condition?

- The same strong dynamics occur in the Noah model at the lower boundary condition at many sites whereas, in this layer, POME and SCAN data better correspond to each other

- In some cases, the SCAN data sometimes show strong (or strange) soil moisture dynamics in the lower part of the profile (2013, 2115) while dynamics in the upper part of the profile is much less pronounced. Please explain.

Specific comments and technical corrections:

- L5: please correct: ALEXI
- L39: replace "and" by "that"
- L43: define CONUS

L67-84: In my view, these two paragraphs would better fit into the methods section

L153: typo: consistent

L159: yes – please define which meteorological forcing was applied in Noah. Was the model calibrated and if yes, to which data?
L162-165: Yes but then it must be proven that the POME-derived soil moisture profiles are correct. How will they be validated if not in a similar approach as applied in this manuscript?

L170: Please provide reference for the MW soil moisture depth? I think at X-band it should be even less than 2-2.5 cm

L183-185: No sentence - please revise

- L337: How were the POME based profiles aggregated?
- L424: Don't the RS measurements "see" the wet surfaces caused by irrigation?
- L 433: please correct: Fig. 4
- L 459: How was this aggregation done?

L482: Aren't the SCAN observations the "true" SM? The RS soil moisture contents are also estimates.

---

## Author Comment (AC2) · 7 Dec 2017

**General Comments:**

1. AMSR-E downscaling: I would first suggest the authors to clarify their research goals. If providing high resolution data is not part of the goal, the authors can perform analysis at 25-km, i.e. upscale ALEXI. Downscaling MS is usually challenging. The consequences on POME are also difficult to evaluate, as pointed by the authors (page 16, line 535).

**Response:** One of the primary goals of the research is to provide remotely sensed soil moisture profiles at operational or near operational (1-5 km), spatial resolutions. At present the authors have a suite of hydrologic and agricultural models running over the Southeastern United States at a 4.7-km resolution to be consistent with the NWS Stage IV multi-sensor precipitation product. The NASA SPoRT project at MSFC also maintains a number of near-real-time land surface models within the NASA Land Information System (LIS) at a 3-km resolution (SPoRT LIS). So the purpose of our research was to provide soil moisture profiles at a resolution that would be consistent with our current modeling system and that can be compared to the land surface models within the SPoRT-LIS framework. We also note that the downscaling method we utilized appears to have improved the coarse resolution microwave data in a number of ways. We will revise the objectives section of the manuscript to provide further explanation of our goals in this regard (see response to detailed comment #2 below).

2. The proposed method can only handle cases with soil moisture is linearly increasing/decreasing with depth, if I am correct. If that is the case, the authors should discuss why the proposed method is preferable than other remote sensing based method, e.g. exponential filter (Albergel et al., 2008).

**Response:** We thank the referee for this comment and agree that perhaps the manuscript can be improved by further discussion on this point. While the entropy method does assume a uniform probability distribution of soil moisture initially, the final profiles are not linear in real space after the integral is performed subject to the boundary conditions and mean moisture content. We refer the referee to our previous paper on this point (Mishra et al., 2013) where a full range of profiles are shown corresponding to all possible cases that may arise in nature. Unlike other methods, the entropy approach suffers from no a priori assumptions about the nature or shape of the moisture profiles in real space. The method is a statistical approach and guarantees the minimum variance unbiased profile subject to the boundary and initial conditions specified. We consider this an improvement over other analytical methods that do presuppose a functional form of the soil moisture distribution. We refer the referee to the article by Singh, (2010) for a full explanation of the theory of entropy of moisture movement in porous media. Further, the method can easily be modified even to include non-monotonic profiles through the identification of the inflexion point based on a few well-known principles of vadose zone hydrology (Mishra et al., 2015). In addition, the proposed method is ideal for the integration of remotely sensed data from multiple sensors which is one of the primary goals of the research. In fact, the primary objective of the project was to merge

microwave (MW) and thermal infrared (TIR) soil moisture estimates into a unified profile. We will revise the manuscript to add this material.

3. Please add units to all the figures

**Response:** Thank you for pointing out this oversight.

4. The conclusions should be presented in a more concisely.

**Response:** We will endeavor to condense the conclusion section by converting bullets into 3 concise paragraphs. We will then add another paragraph with condensed version of the error issues.

**Detailed comments:**

1. Line 16 to 18: please add units to all the numbers being reported. I assume it is in m3/m3.

Response: Thank you for pointing out this oversight.

2. Line 67 to 69: please revise/modify the goal here. The authors should at least mention the methodology should satisfy what applications.

**Response:** If we understand the comment correctly, the referee would like for us to mention the purpose, or potential uses if the developed profiles here. As the referee mentions in the first item above, perhaps this is a good place to position some of the material discussed in that item.

3. Section 2.1: Please specify why this area is selected. It is known that AMSR-E has the poorest performances over dense vegetation areas. This means AMSR-E is usually more accurate over southwest part of the CONUS and less accurate over the eastern part of the CONUS.

**Response:** This is an excellent point. As mentioned previously, the authors currently participate in an extensive research program centered in the Southeastern US. The region is data rich, thus providing us with ample opportunities to test and validate the results. However, as the referee mentions, the region represents a humid subtropical climate with nearly 85% of the area represented by either forest or shrubs, making it one of the toughest regions for MW based SM estimations. As pointed out in the manuscript, the X-band MW signal is greatly attenuated by moderate to high vegetation cover such as represented by the selected study region. Therefore, we feel that, in addition to providing added benefit to the research that we are already doing in the Southeast, the study represents an opportunity to evaluate the performance of the merged MW/TIR profiles in a challenging environment. If successful, then the study will provide greater confidence

towards the robustness of the method. We believe that the results of the study do provide evidence of that robustness. This will be clarified in the revised manuscript.

4. Section 2.2.1: Please justify why LPRM based C-band AMSR-E data were not used, since it is usually considered to have a better quality?

**Response:** It is known that the lower frequency bands are better for surface SM detection since the higher bands suffer disproportionately from the effects of atmospheric interference, vegetation cover and radio interference (Albergel et al., 2011; Brocca et al., 2011). This study employs one of the more extensively used and validated MW based SM data sets from the AMSR-E (2002-2011) mission operating in the X-band frequency from the National Snow and Ice Data Center (NSIDC) and employs the standard NASA retrieval algorithm (Njoku et al., 2003). The NSIDC is one of two AMSR-E data sets supported by NASA with the other being the Vrije Universiteit Amsterdam - Land Parameter Retrieval Model (VUA-LPRM) data. The LPRM uses a single dual polarized channel (X- or C- band) to deduce relationships between geophysical variables such as soil moisture and vegetation characteristics and brightness temperatures (Cho et al., 2015). Several studies such as Wagner et al. (2007); Draper et al. (2011); Jackson et al. (2010); Gruhier et al. (2008) etc. have compared the two data sets and the general conclusion has been that the VUA-LPRM algorithm may be slightly superior in terms of correlation to *in-situ* data, especially at lower latitudes and sparse vegetation (Brocca et al., 2011). However, Jackson et al. (2010) found that for southeast U.S, the NASA retrieval algorithm outperformed the LPRM in terms of bias and RMSE. Furthermore, as pointed out by authors such as Njoku et al. (2005); Jackson et al. (2010) the effects of radio frequency interference (RFI) on C-band are more pronounced over countries such as United States and Japan, therefore X-band retrievals are preferred over such regions. Hence X-band based on the standard NASA (or NSIDC) data set was selected for this study. This issue will be further discussed in the revised manuscript.

5. Equation 6 and 7: the author can remove one of them

**Response:** Thank you. We will remove equations 6 and 7.

6. The captions of figure 2 should be modified. Please specify which products are compared in the caption.

**Response: We thank the referee for pointing out this oversight.**

7. Line 343 and elsewhere: RMSE is the root mean square error. It can be calculated when you have a known "truth" or a very good reference. If NOAH is not assumed to be perfect, I would suggest the author to change it into RMSD, i.e. root mean square difference.

**Response:** We agree with the reviewer, since neither NOAH nor *in-situ* observations can be considered as a perfect reference due to their inherit errors and scale mismatch issues, we will rectify this mistake and replace RMSE with RMSD in the manuscript.

8. Line 349: provide the unit here is m3/m3, I would not say ubRMSE = 0.06 m3/m3 is small. . .

**Response:** We agree that RMSE/RMSD =  $0.06 \text{ m}^3/\text{m}^3$  is little on the higher side, however the statement in question (Line 349) in quotes, "*Moreover < 97% pixels across the study area showed ubRMSE of less than 0.06 across all layers, indicating good agreement between the POME model and Noah SM estimates*". The greater than sign was wrongly put in as less than. The statement simply intent to states that the for more than 97% of pixels the ubRMSE was less than 0.06 m3/m3

Further, as pointed out by Jackson et al. (2010), the agencies such as NASA and JAXA specified an accuracy goal of errors less than  $0.06 \text{ m}^3/\text{m}^3$  for AMSR-E retrievals. Since this study utilizes AMSR-E SM estimates, we assumed the similar accuracy standard. Although, we acknowledge some inconsistencies, such as spatial mismatch (originally AMSR-E spatial resolution is 25 km while we downscaled it to 5 km) and the specified accuracy goal was limited to surface measurements so we assumed it would be appropriate to adopt it for the rootzone as well.

9. Line 353: the author may need to define a threshold of "well" or "good". As shown in the third row of Figure 2, large fractions of correlations are below 0.4. It is hardly to be considered as "well" in my background. However, I agree that this threshold varies according to different applications.

**Response:** We agree that correlations below 0.4 would not be generally considered "good". In this paragraph our main point was that the better correlations (0.6 and above) were generally in the agricultural areas. We think that most people would consider such correlations "good" when dealing with remotely sensed data. Further we characterize correlations around 0.5 (0.46-0.54) as "fairly" good, a designation we think many people would agree with. While the figure does show that there are areas where the correlations are not as good, the discussion is focused on the overall correlations.

10. please add row indices.

**Response:** We assume that the referee is referring to the identifications on the rows (i.e., 2009. 2114, etc.). These are the SCAN station ID's and we will add that to the figure.

11. Line 476 to 477: please rephrase.

**Response:** We assume that the referee is referring to the sentence that begins "Figure 6 shows that ...". Perhaps the objection is to the general characterization that the figure is showing the poorly defined "accuracy" of the method. If this is the case, then we apologize for the poor wording here. We will expand this section to be more specific in what we mean by accuracy according to the indices reflected in the figure.

12. Line 478: the implementation of TC should include more information. Was the climatology removed from each dataset?

**Response:** In this study, we used covariance notation to compute TC error, which allows us to solve for the unscaled error variances directly. As pointed out by Gruber et al., (2016), with covariance notation, the posterior scaling of dataset is possible and is an optional process. Also, both scaled based difference notation as well as unscaled covariance notation mathematically leads to same results, therefore either method can be used. Further, the advantage of using covariance notation is that it provides an estimate of sensitivity of the dataset to soil moisture change. The sensitivity estimates allows for additional validation and inter-comparison of the datasets. We will include this information in the revised manuscript.

13. Line 484 and figure 7: it is not common to express TC results in R2. Please specify how this metric was derived.

**Response:** Thanks for pointing this out. As mentioned in response 12 above, the study uses covariance notation for TC analysis which also estimates sensitivity of the datasets represented as:  $\beta_i^2 \sigma_{\theta}^2$ , which can be used to further validate and inter-compare datasets. Recently, McColl et al., (2014) proposed to use TC analysis to estimate the correlation coefficient between datasets involved and the underlying 'true' signal as:

$$R_i^2 = \frac{\beta_i^2 \sigma_{\theta}^2}{\beta_i^2 \sigma_{\theta}^2 + \sigma_{\varepsilon_i}^2}$$
$$\beta_X^2 \sigma_{\theta}^2 = \frac{\sigma_{XY} \sigma_{XZ}}{\sigma_{YZ}}$$

Where X,Y,Z refers to three datasets involved in TC analysis.

14. Line 488 to 490 is incorrect. TC estimates the total error, instead of just the random error. Please see Yilmaz and Crow 2014. This means if NLDAS is less accurate, either due to random or temporally correlated errors, it will be shown in the TC results.

**Response:** We agree that TC includes total error and did not mean to infer otherwise. We apologize for the confusion. The point we were making is that the root zone soil moisture values in the NOAH LSM are the result of a deterministic equation and thus are not observations of a random variable or statistical estimates as in the POME method. For this reason, we were a little uncomfortable in making decisive statements with regard to the behavior of the NOAH data. We will rephrase this paragraph to make clear our intentions, or if the editors prefer we can remove it altogether.

15. Section 4.4.1: please refer to my general comment. If the reviewer can perform analysis at coarse scales, this section is unnecessary. This may make the manuscript cleaner.

**Response:** Please refer to response 1

16. Line 578 to 579: I would not consider the minimum bias is the key advantage of POME. This is because it is nearly impossible to define an absolute bias at large scales, since the reference dataset (e.g. SCAN) can also be biased.

**Response:** We thank the referee for this observation and agree that there is bias in the SCAN data. Here we are merely comparing the bias of the different approaches. We agree that because of scale disparities we cannot make definitive statements relative to true bias. We will revise this section accordingly to acknowledge that bias exists in all three data sets.

17. Line 585: Root zone soil moisture at large scales can have significant spatial variability, according to my experience. It can result in large errors/bias using limited point sensors to represent large scale root zone soil moisture. Hence, I'm suspecting how confidently the authors can draw this conclusion.

**Response:** We agree with this statement to a large extent. We are certainly aware of the uncertainties in using point data to represent larger spatial domains. In fact, this is the reason that we added the gridded NOAH soil moisture estimates as a comparison metric in addition to the point SCAN data. However, in our review of previous studies such as ours we found that it was not uncommon for other authors to use point data as we did (we cited several of these in the manuscript). To counter this, we added the gridded data sets and the TC analysis where the point data are treated as merely another estimate of the true value of the grid soil moisture. We feel that the results of all the analyses are compatible and lead to similar conclusions.

18. Line 598 to 600: please refer to my General Comment 1

**Response:** Please refer to response 1

**References:**

- Albergel, C., Zakharova, E., Calvet, J.-C., Zribi, M., Pardé, M., Wigneron, J.-P., Novello, N., Kerr, Y., Mialon, A., Fritz, N.-D., 2011. A first assessment of the SMOS data in southwestern France using in situ and airborne soil moisture estimates: The CAROLS airborne campaign. Remote Sens. Environ. 115, 2718–2728. doi:10.1016/j.rse.2011.06.012
- Brocca, L., Hasenauer, S., Lacava, T., Melone, F., Moramarco, T., Wagner, W., Dorigo, W., Matgen, P., Martínez-Fernández, J., Llorens, P., Latron, J., Martin, C., Bittelli, M., 2011. Soil moisture estimation through ASCAT and AMSR-E sensors: An intercomparison and validation study across Europe. Remote Sens. Environ. 115, 3390– 3408. doi:10.1016/j.rse.2011.08.003
- Cho, E., Choi, M., Wagner, W., 2015. An assessment of remotely sensed surface and root zone soil moisture through active and passive sensors in northeast Asia. Remote Sens. Environ. 160, 166–179. doi:10.1016/j.rse.2015.01.013
- Draper, C., Mahfouf, J.F., Calvet, J.C., Martin, E., Wagner, W., 2011. Assimilation of ASCAT near-surface soil moisture into the SIM hydrological model over France. Hydrol. Earth Syst. Sci. 15, 3829–3841. doi:10.5194/hess-15-3829-2011

Gruber, A., Su, C., Zwieback, S., Crow, W., Dorigo, W., Wagner, W., 2016. Recent advances in (soil moisture) triple

collocation analysis. Int. J. Appl. Earth Obs. Geoinf. 45, 200–211.

- Gruhier, C., de Rosnay, P., Kerr, Y., Mougin, E., Ceschia, E., Calvet, J.-C., Richaume, P., 2008. Evaluation of AMSR-E soil moisture product based on ground measurements over temperate and semi-arid regions. Geophys. Res. Lett. 35, L10405. doi:10.1029/2008GL033330
- Jackson, T.J., Cosh, M.H., Bindlish, R., Starks, P.J., Bosch, D.D., Seyfried, M., Goodrich, D.C., Moran, M.S., Du, J., Goodrich, D.C., Moran, M.S., 2010. Validation of Advanced Microwave Scanning Radiometer Soil Moisture Products. IEEE Trans. Geosci. Remote Sens. 48. doi:10.1109/TGRS.2010.2051035
- McColl, K.A., Vogelzang, J., Konings, A.G., Entekhabi, D., Piles, M., Stoffelen, A., 2014. Extended triple collocation: Estimating errors and correlation coefficients with respect to an unknown target. Geophys. Res. Lett. 41, 6229–6236. doi:10.1002/2014GL061322
- Mishra, V., Cruise, J., Mecikalski, J., Hain, C., Anderson, M., 2013. A Remote-Sensing Driven Tool for Estimating Crop Stress and Yields. Remote Sens. 5, 3331–3356. doi:10.3390/rs5073331
- Mishra, V., Ellenburg, W., Al-Hamdan, O., Bruce, J., Cruise, J., 2015. Modeling Soil Moisture Profiles in Irrigated Fields by the Principle of Maximum Entropy. Entropy 17, 4454–4484. doi:10.3390/e17064454
- Njoku, E.G., Ashcroft, P., Chan, T.K., Li Li, 2005. Global survey and statistics of radio-frequency interference in AMSR-E land observations. IEEE Trans. Geosci. Remote Sens. 43, 938–947. doi:10.1109/TGRS.2004.837507
- Njoku, E.G., Jackson, T.J., Lakshmi, V., Chan, T.K., Nghiem, S. V., 2003. Soil moisture retrieval from AMSR-E. IEEE Trans. Geosci. Remote Sens. 41, 215–228. doi:10.1109/TGRS.2002.808243
- Singh, V.P., 2010. Entropy theory for movement of moisture in soils. Water Resour. Res. 46, n/a–n/a. doi:10.1029/2009WR008288
- Wagner, W., Blöschl, G., Pampaloni, P., Calvet, J.-C., Bizzarri, B., Wigneron, J.-P., Kerr, Y., 2007. Operational readiness of microwave remote sensing of soil moisture for hydrologic applications. Hydrol. Res. 38.

---

## Author Comment (AC3) · 7 Dec 2017

POME model: I would still like to have some more information about the model. It only requires surface soil moisture and soil profile mean soil moisture as input information and assumes that moisture is either monotonically increasing or decreasing. If this is not the case, an inflection point that is located in the soil layer with the highest field capacity is assumed

1. How does the POME model consider several layers of different texture? They might exhibit rather different soil moisture contents with distinct "jumps" between layers. In addition, infiltration fronts might induce further "inflection points". I cannot really imagine how this can be estimated from surface soil moisture and mean profile water con-tent only. Will there be just one optimal soil moisture profile or might there be several (concept of equifinality).

**Response:** The referee makes an excellent point about the behavior of soil moisture in real world situations. First, the POME model uses effective SM to develop a profile (eq. 4, Line 280), the boundary conditions and mean are first computed in effective SM and the model is applied. Finally, the developed profile in effective SM is converted to volumetric soil moisture values based on layer soil texture. However, one assumption made here is that the mean effective SM (available from the ALEXI model) is a composite of all the layers with different textures.

Now, as to the observation made by the reviewer, profiles can certainly be very irregular due to the different characteristics of the multiple layers that make up the soil column. As discussed in the manuscript and in the earlier paper by Mishra et al., (2015), the POME method as currently formulated envisions the profile as either monotonically increasing or decreasing or as possessing a prominent inflexion point. However, the method is a statistical procedure that ensures the minimum variance unbiased profile given the input data. In short, this ensures that the profile will be the best fit possible given what we know *i.e.*, the surface and lower boundary values and the mean. As shown in multiple earlier studies (Al-Hamdan and Cruise, 2010; Mishra et al., 2015, 2013; Singh, 2010a, 2010b) the resulting profile will normally be the best fit line through irregular points of a natural profile if the input data are approximately correct. We believe that this is the best any analytical method can achieve. We will add an appendix with more description of the POME model.

**2. How is the lower boundary condition parameterized?**

**Response:** We consider the lower boundary to potentially be a calibration parameter. There are various ways it can be estimated. For example, it could be considered a soils parameter or it can be used to link the climatology of the POME profile to a land surface model. In this study, the lower boundary was merely parameterized at 50% of available water content. As shown in past analyses (e.g., Mishra et al. (2013)) moisture content in the deeper layers is often fairly constant over the year, depending on the soil texture, and can thus can sometimes be set as the lower boundary on SM. We felt since the present study was a proof of concept in many ways, that parameterizing the lower boundary in this way was appropriate.

As the authors state correctly, both the Noah and the POME model will be affected by model errors and consequently may provide erroneous soil moisture profiles while the in situ data were measured at a much smaller measurement scale and the measurement site may not be representative for the entire pixel. Nevertheless, for a soil hydrologist, the time series shown in Figure 6 provide the most valuable information for identifying model errors and weaknesses of models and data. Unfortunately, this figure is almost not discussed within the manuscript. Observations to be addressed are e.g.

3. Except some sites where the SCAN data show some bias (mostly towards the dry range), SCAN and NOAH data appear to agree rather well, both in absolute soil moisture content and in the amplitude of the annual dynamics whereas POME often shows a very strong dynamics (especially towards the wet range). Might this cause problem with the upper boundary condition?

**Response:** We agree with the referee that we did not make sufficient use of the SM series shown in Figure 5. We thought that the salient points could be made through the statistical analyses and ignored the fact that perhaps the issues could be better illuminated through the figure itself. We assume that the referee is referring in this comment primarily to stations 2027, 2053, 2078, 2037, and 2013 where the POME SM estimates show a pronounced cycling effect in the upper layer not evident in the SCAN or NOAH data. We note here that the surface SM from POME is primarily set by the boundary condition provided by the MW data and that the overall correlation among all stations in this layer was about 0.55. However, as the referee states, those particular 5 stations showed anomalies that were not in sync with either the SCAN data or the NOAH simulations. From Figure 1 and Table 1 it can be observed that all of these stations are located in agricultural or mixed crop land coverages. In addition, the stations in North Alabama and southwest Georgia are in heavily irrigated areas. Obviously, the SCAN station would not be irrigated and the NOAH model also does not include it. The ability of the microwave instrument to sense the moisture from irrigation accounts for the seasonal cycle evident in the POME profiles at those sites. In these cases, it is probable that the POME profiles are more accurate over the 5-km spatial grid than are the SCAN or NOAH data. We failed to make this point anywhere in the manuscript and we greatly appreciate the referee pointing it out to us.

Further, the other two sites are located in areas where quite a bit of surface water is present in the remotely sensed pixels. It is possible that the seasonal cycle of water levels in the area account for the behavior seen at these locations.

We will revise the manuscript to add the above discussion to Section 4.3 where Figure 5 is first raised. The later statistical analyses can then be used to support the observations raised there.

4. The same strong dynamics occur in the Noah model at the lower boundary condition at many sites whereas, in this layer, POME and SCAN data better correspond to each other.

**Response:** We completely agree and as mentioned several times throughout the manuscript, this observation provides further evidence of how the POME model tends to improve relative to the SCAN data in the deeper layers of the soil column. It is also evidence that for most of the SCAN

sites, the lower boundary condition (at 50% of available water content) seems to be a reasonable assumption for the POME model.

5. In some cases, the SCAN data sometimes show strong (or strange) soil moisture dynamics in the lower part of the profile (2013, 2115) while dynamics in the upper part of the profile is much less pronounced. Please explain.

**Response:** We too were struck by the behavior at these two sites and conducted an extensive investigation. The SCAN site SM at these locations showed highly seasonal dynamics at the lower depths. Site 2115 is characterized mostly as sandy soil from 0-45 cm depth. These soils have a much lower water holding capacity that leads to increased infiltration to the lower depths. Having sandy soils in the upper layers in part can explain the seasonal variations at the lower depths. Similarly, at site 2013 the upper two layers (0-33 cm) are recognized as sandy loam although the lower layer at the SCAN site exhibits high clay content.

**Specific comments and technical corrections:**

- L5: please correct: ALEXI Response: Thank you for pointing out this mistake.
- 2. L39: replace "and" by "that" **Response**: We appreciate the correction.
- 3. L43: define CONUS **Response:** Thank you. We will make this correction
- 4. L67-84: In my view, these two paragraphs would better fit into the methods section **Response:** We placed this material at the end of the Introduction to define the objectives of the study and to provide a brief overview of what was done. If the editors think it appropriate, we are certainly open to removing it from this section.
- 5. L153: typo: consistent **Response:** Thank you. We will make this correction.
- 6. L159: yes please define which meteorological forcing was applied in Noah. Was the model calibrated and if yes, to which data? **Response:** A real-time version of the Land Information System (LIS) has been maintained by NASA Short-term Prediction Research and Transition Center (SPoRT) for use in experimental operations by both domestic and international operational weather forecasters (Case et al., 2016, 2008). The basis of the SPoRT-LIS is a 33-year soil moisture climatology simulation spanning 1981–2013 and extended to the present time, forced by atmospheric analyses from the operational North American Land Data Assimilation System-Phase 2 (Xia et al., 2012). We will clarify this in the revised manuscript.

7. L162-165: Yes but then it must be proven that the POME-derived soil moisture profiles are correct. How will they be validated if not in a similar approach as applied in this manuscript?

**Response:** We appreciate this point. The POME approach has been applied under strictly controlled laboratory conditions (Al-Hamdan and Cruise, 2010; Singh, 2010a), field scale applications (Mishra et al., 2013) and compared to a detailed mathematical model of soil moisture movement (Mishra et al., 2015). In all of these instances, the method has been shown to be highly accurate (error on the order of 3%) when the input data are known accurately. In addition, entropy theory (Jaynes, 1957) indicates that the method will yield the optimal profile subject to the input data.

- L170: Please provide reference for the MW soil moisture depth? I think at X-band it should be even less than 2-2.5 cm
  Response: Thank you for pointing out this oversight. The penetration depth of the X-band microwaves is indeed 0-2 cm.
- 9. L183-185: No sentence Response: Thank you for pointing it out, it will be replaced by "A time differential application of the ALEXI model was performed to monitor the rise in land surface temperature (LST) from morning to local noon. The early-day rise in LST is used to diagnose the partitioning of net radiation into sensible; latent and soil heat fluxes."
- L337: How were the POME based profiles aggregated? Response: The initial POME profiles were generated at 5 cm layer depths. A simple unweighted (as all layers being aggregated are of equal depth) mean was used to aggregate the SM values from 5 cm layer depths to represent a layer depth consistent with Noah LSM layer depths.
- 11. L424: Don't the RS measurements "see" the wet surfaces caused by irrigation? **Response:** Exactly, the remotely sensed data are picking up the added water from the irrigation in this area whereas the SCAN site (which is not located in a private irrigated field) is not "seeing" this water and likewise it is not accounted for in the NOAH model. Thus, the positive bias here is not necessarily bad.
- 12. L 433: please correct: Fig. 4Response: Thanks for pointing that out. We will make the correction.
- 13. L 459: How was this aggregation done? **Response:** Please refer to response 10
- 14. L482: Aren't the SCAN observations the "true" SM? The RS soil moisture contents are also estimates.
  **Response:** Thank you for the comment. In triple collocation (TC) analysis the three datasets (including the point SCAN observations) are considered as three independent estimates of the true average soil moisture condition of the pixel. In this analysis there is

no way to know the "true" soil moisture. Rather the error in the three datasets is computed relative to the unknown "truth" so that they can be compared to one another.

**References:**

- Al-Hamdan, O.Z., Cruise, J.F., 2010. Soil Moisture Profile Development from Surface Observations by Principle of Maximum Entropy. J. Hydrol. Eng. 15, 327–337. doi:10.1061/(ASCE)HE.1943-5584.0000196
- Case, J.L., Crosson, W.L., Kumar, S. V., Lapenta, W.M., Peters-Lidard, C.D., 2008. Impacts of High-Resolution Land Surface Initialization on Regional Sensible Weather Forecasts from the WRF Model. J. Hydrometeorol. 9, 1249–1266. doi:10.1175/2008JHM990.1
- Case, J.L., Mungai, J., Sakwa, V., Zavodsky, B.T., Srikishen, J., Limaye, A., Blankenship, C.B., 2016. Transitioning Enhanced Land Surface Initialization and Model Verification Capabilities to the Kenya Meteorological Department (KMD).
- Jaynes, E.T., 1957. Information Theory and Statistical Mechanics I. Phys. Rev. 108, 171–190.
- Mishra, V., Cruise, J., Mecikalski, J., Hain, C., Anderson, M., 2013. A Remote-Sensing Driven Tool for Estimating Crop Stress and Yields. Remote Sens. 5, 3331–3356. doi:10.3390/rs5073331
- Mishra, V., Ellenburg, W., Al-Hamdan, O., Bruce, J., Cruise, J., 2015. Modeling Soil Moisture Profiles in Irrigated Fields by the Principle of Maximum Entropy. Entropy 17, 4454–4484. doi:10.3390/e17064454
- Singh, V.P., 2010a. Entropy theory for movement of moisture in soils. Water Resour. Res. 46, n/a–n/a. doi:10.1029/2009WR008288
- Singh, V.P., 2010b. Entropy theory for derivation of infiltration equations. Water Resour. Res. doi:10.1029/2009WR008193
- Xia, Y., Mitchell, K., Ek, M., Sheffield, J., Cosgrove, B., Wood, E., Luo, L., Alonge, C., Wei, H., Meng, J., Livneh, B., Lettenmaier, D., Koren, V., Duan, Q., Mo, K., Fan, Y., Mocko, D., 2012. Continental-scale water and energy flux analysis and validation for the North American Land Data Assimilation System project phase 2 (NLDAS-2): 1. Intercomparison and application of model products. J. Geophys. Res. Atmos. 117, n/a–n/a. doi:10.1029/2011JD016048

---

## Author Response (AR2)

Dear Authors,

I am happy to say that all three referees see substantial improvements in your manuscript. They are also clear in that your study is covering a challenging topic that is of great interest to the community. However, the referees also state that the manuscript would profit from a number of improvements and clarifications and have made some very constructive comments. In the light of providing the community with a clear and convincing publication on this topic I recommend to make another effort in revising your manuscript according to the referees' suggestions and by providing answers to the still open questions identified by the referees.

I am looking forward to your revised manuscript.
Best regards,
Theresa Blume

**Anonymous during peer-review:** Yes **No**
**Anonymous in acknowledgements of published article:** Yes **No**

**Recommendation to the Editor**

| | |
|---|---|
| **1) Scientific Significance**
Does the manuscript represent a substantial contribution to scientific progress within the scope of this journal (substantial new concepts, ideas, methods, or data)? | Excellent **Good** Fair Poor |
| **2) Scientific Quality**
Are the scientific approach and applied methods valid? Are the results discussed in an appropriate and balanced way (consideration of related work, including appropriate references)? | Excellent **Good** Fair Poor |
| **3) Presentation Quality**
Are the scientific results and conclusions presented in a clear, concise, and well structured way (number and quality of figures/tables, appropriate use of English language)? | Excellent **Good** Fair Poor |

For final publication, the manuscript should be

**accepted as is**

accepted subject to **technical corrections**

accepted subject to **minor revisions**

**reconsidered after major revisions**

    **I am willing to review the revised paper.**

    I am **not** willing to review the revised paper.

**Rejected**

**Suggestions for revision or reasons for rejection (will be published if the paper is**

**accepted for final publication)**

The manuscript introduces a challenging research topic still unsolved.
The manuscript is well organized and authors have put great effort on the review of the manuscript. I had the opportunity to read both the manuscript, the rebuttal letter and also the manuscript with track of changes. I have to admit that the manuscript has been improved significantly, but according to my perspective there are still a number of open questions.

I have some comments that have been written with the aim to improve the final quality of the manuscript:

1. The authors use the relationship between relative saturation and evapotranspiration to derive an estimate of the available water content. I would like to stress this point because the ET can certainly provide good indications on the available water content, but when the relative saturation exceeds the field capacity, this method is unable to properly identify the state of the soil. Therefore, ALEXI SM is an estimate of the available water content not soil moisture.
**Response:** We agree with the reviewer that the relationship between fractional ET and available water content has limitations. It can only effectively provide SM information between wilting point and field capacity. However, since ALEXI senses the moisture content of the entire root-zone (~0-100 cm), it would require a very large rain event for the entire soil column to experience a moisture content greater than field capacity, or a prolonged drought to produce SM less than wilting point. Further, if conditions above field capacity are underestimated, then the root zone may dry out faster than observed, but theoretically that would be corrected with the next available retrieval. Therefore if ALEXI retrieval is available within next few (~2-4) days the errors would be rather small (page 7 lines 11-15).

2. Given the above consideration I wonder if such limitation may be reasonable of some of the errors observed may be due to such an assumption.
**Response:** We agree with the reviewer that the empirical relationship between fraction ET and available water content has some limitations and may be responsible for some errors in the POME profiles. This point is further clarified in the revised MS page 7 lines 12-15.

3. Looking at the results, it seems that the proposed model leads to results similar to those by Noah LSM. The argument of the authors is that POME tends to be better in the lower soil layer, but this result is imposed by calibrating the lower limit of relative saturation profile. As stated by the authors the lower boundary condition is imposed equal to 0.55. In fact, all time series in the lower layer (40-100cm) show a similar dynamics with small variability around a relative value close to 0.55. Therefore, this is not an evidence of model reliability in my personal opinion.
**Response:** We agree with the reviewer that the POME profiles are affected by the lower boundary condition which in this case is set to 50% of available water content. First, this approximation is soil type dependent (and therefore not strictly constant), and secondly, it is based on experience that soil moisture in the deeper soil zones varies less dramatically throughout the year due to lower root density (resulting in minimal root water uptake) and dampened connectivity to variations in the surface layer. Since in many ways this is a proof-of-concept study, we felt this was appropriate for an initial appraisal of the technique. The POME implementation was not based on any prior knowledge of specific SM in the bottom layers, and in fact one can argue that its use actually weakened the results in this case. It certainly impacted the correlations, as a simple assumption like this would not be expected to correlate with actual observed soil moisture. As discussed in comment 5 below, a few

stations did exhibit considerable variability in the lower layers in contrast to our assumption. So we do not see how our lower boundary assumption unfairly biased the results in our favor, as implied. On the contrary, we feel that our selection of a lower boundary without any prior knowledge, and that even this assumption led to fairly good results, demonstrates the strength of the approach. These points are discussed in the MS page 9 line 27-30 and page 10 lines 1-4.

However, we have mentioned in the manuscript (page 16 line 2-4) that the lower boundary can be parameterized (as done here) or used as a calibration factor. Since the ultimate purpose of the profiles is to be assimilated into a Land Surface Model (LSM) or agricultural model, the lower boundary can also serve as an ideal point to tie the remotely sensed profile to the climatology of the model. In fact, we have done this in a companion paper and found that the POME profile is greatly improved.

4. Referee 2 comment 3. I think that the discrepancies between soil moisture measurements in the top layer and the POME predictions may be due to the ALEXI SM estimates that are probably rescaled according to the relative soil porosity.
**Response:** We appreciate this observation and the referee is correct in that the ALEXI data are functions of the soil properties as stated above. However, in this case, the SM in the top layer of the POME model is a result of the MW estimates used as the surface boundary, not the ALEXI model.

5. In my experience, the soil moisture signal tend to be smoothed moving to the lower layers (see Manfreda et al., HESS - 2014), but the time series 2115, 2113, 2013 (figure 5) show an amplification of soil moisture changes over time. This may be due to the presence of a phreatic surface interfering with the unsaturated zone. I suspect that these point are not coherent with the modeling scheme adopted.
**Response:** We thank the referee for this observation and we too were struck by the behavior of these particular stations. We agree that in our experience too most SM profiles show a damping effect with depth and, in fact, that was the reasoning behind our selection of the lower boundary condition as discussed in Number 3 above. We have investigated these particular stations extensively and have concluded that the SM behavior is site specific. We note that these stations are located in cropland/short vegetation areas and thus we do not believe that perched water would be located within 1-m of the surface. Two sites (2013 and 2115) have very porous soils near the surface (2115 even classified as gravel) with clay layers beneath and the other site (2113) is located very near a stream and is probably influenced by surface water intrusion. Of course, we had this information available beforehand but did not want it to influence our selection of the lower boundary on the profile as discussed in our response to Comment 3 above.

6. In general, I admit that the authors are working on a challenging topic, but the overall presentation of the results does not fully convince about the advantages and potential of the proposed method. The fact that POME is superior to Noah LSM only on 50% of the stations and the fact that the better performances in the lower layers, claimed by the authors as an advantage of the proposed method, may be an outcome of a calibration is a bit weak to justify its use in future applications. I would like to have more arguments about the potential of these model, how and where it can be used.
**Response:** We appreciate the reviewer's comment and acknowledgement of the challenges involved in this study. We also appreciate the opportunity to articulate the positive aspects of the recommended approach.

First, it is true that the POME profiles were clearly superior to the Noah LSM in about half of the cases in terms of Bias and RMSE. However, the results were very comparable in the other 50% of the cases. Only in terms of correlation was the LSM significantly better

and the results clearly indicate that this was a function of the two boundary conditions, and particularly the parameterization of the lower boundary. The Noah LSM is executed within NASA LIS operated by the SPoRT team at Marshall Space Flight Center. Noah is driven by the best high resolution climatological and energy data available in the U.S. and possibly the world. On the other hand, the POME model requires only three inputs, two of which are readily available from RS, while the third can be parameterized as done here or used to tie the profile to the climatology of a LSM or to field conditions if known. We would like to reiterate that we did not calibrate the lower boundary to field conditions in this case, and in fact, merely assumed that 50% of available water capacity remained in the spirit of the Maximum Entropy concept of relying on a minimum of *a-prior* information. So we feel that the fact that the POME profiles were superior to the LSM at half the instances, and were comparable everywhere, is a strong positive for the approach. Second, since the POME model can be used with remote sensing observations, it can be applied anywhere on Earth where vegetation is active, which would be particularly useful in data scarce regions of the world. The POME model is also computationally inexpensive whereas LSM models (such as Noah, VIC etc.), which require multiple inputs (ranging from static inputs like soil type, landcover to dynamic inputs such as weather information etc.). These LSMs also often need local inputs for regional calibration and can be computationally expensive over large regions and at finer scales.

However, if we can achieve accuracies similar to LSMs with significantly fewer input parameters, at large spatial scales with negligible computational resource requirements using the POME model (as shown in this study), then we believe the POME model concept has huge potential. Further, the LSMs usually don't take irrigation activities into account (unless grid scale irrigation maps are available), while the POME model, which is based on satellite SM retrievals of surface as well as root-zone, contains the signals from irrigated fields. Therefore, as discussed in the manuscript several times, POME seems to represent both irrigated and non-irrigated fields and provide more realistic estimations.

**Anonymous during peer-review: Yes** No

**Anonymous in acknowledgements of published article: Yes** No

**Recommendation to the Editor**

| | |
|---|---|
| **1) Scientific Significance**
Does the manuscript represent a substantial contribution to scientific progress within the scope of this journal (substantial new concepts, ideas, methods, or data)? | **Excellent** Good Fair Poor |
| **2) Scientific Quality**
Are the scientific approach and applied methods valid? Are the results discussed in an appropriate and balanced way (consideration of related work, including appropriate references)? | **Excellent** Good Fair Poor |
| **3) Presentation Quality**
Are the scientific results and conclusions presented in a clear, concise, and well structured way (number and quality of figures/tables, appropriate use of English language)? | **Excellent** Good Fair Poor |

For final publication, the manuscript should be
**accepted as is**
accepted subject to **technical corrections**
**accepted subject to minor revisions**
reconsidered after **major revisions**

    I am willing to review the revised paper.
    I am **not** willing to review the revised paper.
**rejected**

**Suggestions for revision or reasons for rejection (will be published if the paper is accepted for final publication)**

The manuscript entitled " Development of Soil Moisture Profiles Through Coupled Microwave-Thermal Infrared Observations in the Southeastern United States" presents interesting and relevant research on the development of vertical soil moisture profiles using POME using remote sensing data. The manuscript is well structured and concise. It has already undergone one round of reviews. My comments are mostly addressing some points which need further clarification. In my opinion these are mostly minor issues and therefore I hope to see this article soon published in HESS.

1) The ALEXI SM is calculated using a linear function based on the available water. Can you argue why you use only one equation for the entire domain and should this not be vegetation depended. For example forest which intercepts the rainfall may require another relationship than grassland which has a lower interception.
**Response:** This is an excellent point, and we thank the referee for bringing this into discussion. We agree that there are multiple possible relations between fractional PET and AWC that could be used (linear, piece-wise linear, logarithmic etc.). These relationships are usually dependent on vegetation (tall tree vs medium vs crop vs grassland), soil texture and location type (water limited vs well-watered) (Anderson et al., 2007). An earlier study by Hain et al. (2009) considered four such relationships and found that the hybrid relationship (linear and non-linear merged) provided the best results for Oklahoma region. However, for large scale applications, linear functions are generally used because: a) less detailed soil and vegetation inputs are required for implementation; and b) their sensitivity to soil moisture is constant (Song et al., 2000). Therefore, a linear model was used in this study as well. This point is mentioned in the MS page 7 lines 1-5.

2) If I am not mistaken you do not mention the penetration depth of the AMSR-E microwave instrument. To my knowledge this can vary depending on the soil moisture conditions. How does this affect your methodology and results?
**Response:** The MW penetration depth is a function of its sensors wavelength, penetration depth increases with wavelength. L-band sensors such as SMAP or SMOS with higher wavelength have penetration depths of 3-5 cm while X and C band sensors like AMSR-E have 0-2 cm penetration depths. For this study, the POME model is applied at 5 cm layer depth therefore the surface observations from AMSR-E (0-2cm) were assumed to be representing 0-5cm. We understand there is a mismatch in depths, although the integral function within the POME model which couples boundary conditions from mean moisture content can compensate for the surface layer depth mismatch. The MW sensing depth is mentioned in MS page 17 line 4.

3) The LST data used in ALEXI is constrained to cloud free conditions. However Figure 5 presents complete timeseries of the POME model. I guess this is described in section 3.4,

but I am surprised that you can gap fill the days with cloud cover by using a simple moving window. Please clarify your method and its limitations.

**Response:** Figure 5 actually shows the SM conditions after every 8-day interval. We have updated the caption to clarify this representation. This was done for simplification/figure clarity and to represent the general trends (instead of daily variabilities). Even though we use data from every 8th day, still some data gaps can be seen with POME time series (blue) few sites (e.g. end of 2006 in sites 2009, 2053, 2078 or mid-2009 for sites 2115 and 2027 etc.).

The gap filling method is more effective in root-zone where SM variabilities are relatively less compared to the surface at daily time-step. Earlier studies such as by (Leng et al., 2017a, 2017b) explored the vegetation and aerodynamic coefficient based gap filling algorithm for satellite derived SM estimations. Despite showing promises, the proposed algorithm requires ancillary datasets that are not part of this study. Further, there is a strong correlation between surface and SM dynamics at lower layers for temporal lags of less than 5-days (Alfieri et al., 2017; Penna et al., 2013). Therefore a 3-day moving window can be used to fill in some of the gaps in ALEXI retrievals. This point is further clarified in the revised MS (page-10 lines 9-14)

4) As I understand POME, it is a statistical model not a physically based approach. How can POME account for different soil types which will affect the vertical SM profile?

**Response:** The POME integrals are applied over effective SM conditions. At the time of application, both mean and lower boundary condition data are available in effective SM state (with no prior information regarding soil type), only the surface volumetric SM conditions are known which are converted to effective. After the model application the effective SM profile distribution is then converted to volumetric by mapping the layers soil characteristics information available (Page 9 line 12).

**Anonymous during peer-review: Yes** No

**Anonymous in acknowledgements of published article: Yes** No

**Recommendation to the Editor**

| | |
|---|---|
| **1) Scientific Significance**
Does the manuscript represent a substantial contribution to scientific progress within the scope of this journal (substantial new concepts, ideas, methods, or data)? | **Excellent** Good Fair Poor |
| **2) Scientific Quality**
Are the scientific approach and applied methods valid? Are the results discussed in an appropriate and balanced way (consideration of related work, including appropriate references)? | Excellent **Good** Fair Poor |
| **3) Presentation Quality**
Are the scientific results and conclusions presented in a clear, concise, and well structured way (number and quality of figures/tables, appropriate use of English language)? | Excellent **Good** Fair Poor |

For final publication, the manuscript should be

**accepted as is**

accepted subject to **technical corrections**

**accepted subject to minor revisions**

reconsidered after **major revisions**

    I am willing to review the revised paper.

    I am **not** willing to review the revised paper.

**Rejected**

**Suggestions for revision or reasons for rejection (will be published if the paper is accepted for final publication)**

Review HESS-2017-351

First of all I would like to make clear that I have not been a reviewer on this manuscript before. I studied the response to the reviewers and the revised manuscript and I think the authors responded well to most of these comments, but being a new reviewer for this manuscript I cannot avoid having some questions. I found the manuscript extremely interesting, and the approach without a-priori assumptions at the scale of LSM models is very useful to the community of HESS. At present however, the manuscript is somewhat impenetrable to me, and therefore I assume to other readers of HESS as well (comments below). Which readership did the authors have in mind: the well-informed LSM or remote sensing community that is certainly present in HESS, or the broader hydrologic community with an interest in remotely sensed applications?

1. The objective of the paper evolves around the POME model by using SM profiles developed from remotely sensed data only. Although I am sure everyone knows about entropy, the explanation of POME is rather short, as mentioned by one other reviewer as well. The reader is referred to several previous publications that employ the POME model. I suggest to include a more extensive description of the POME results instead of referring to whom applied it. This would help out the interested non-expert HESS readership to get to a level of insight that helps to appreciate this manuscript.

**Response:** We thank the reviewer for the comment. In the earlier revision, we included more details on the POME model and its applications. However, given the length of the paper we were little conservative in the model description. But in the current revision we have added an appendix-A detailing the POME model description. We agree with the reviewer that having a detailed POME model description will definitely add more value to the manuscript and will be more interesting to the readers.

2. For ALEXI field capacity and wilting point are employed in Eq. 1. Which values were used for the different soil types?

**Response:** We thank the reviewer for raising this point. We have added a table listing all the major soil types of the region with their characteristics values used in this study as an appendix B.

3. Line 250-253: Can the authors explain a bit more about the varying spatial scales and locations instead of only giving r and RMSE/RMSD/ubRMSD for various authors? For scientists unfamiliar with SEE disaggregation this part could be more informative, also because it is important for the approach used in this study.

**Response:** We agree with the reviewer, adding information about spatial resolutions will provide the reader with a more complete description of the downscaling approach. We have revised the MS to add the spatial resolution information as well as included the citation of our recent publication solely on downscaling approach using ALEXI surface evaporation over Continental U.S. Please refer to page-7 lines 22-25; page-8 lines 29-30 and page-9 lines 1-2 of the revised MS.

4. What was the rationale/advantage for using surface and potential evaporation instead of

the SEE definition given in line 275? Not clear now, only clear it changes the definition. Would be good to know as the change in definition seems to limit its applicability to water rich environments?

**Response:** We thank the reviewer for raising this point. The SEE definition provided in Line 275 (MS rev1) refers to the original approach used by Merlin et al. (2013, 2012) where SEE is computed using MODIS land surface temperature, vegetation index and albedo information. This was an indirect approach to estimate the ratio of surface evaporation to potential surface evaporation (i.e. SEE). In this study, however, the SEE is computed directly from actual surface evaporation and potential surface evaporation data. Conceptually, the SEE employed by Merlin, et al and in this study is the same. The surface evaporation and potential surface evaporation data were available relatively easily since the former is coming from the ALEXI model, which is already an important part of this study. Furthermore, for consistency purposes the utilization of same data source (ALEXI) for surface SM disaggregation and root-zone mean moisture content seems a better approach.

The issue of water rich vs dry environment arises with the selection of $\frac{\partial SM_{mod}}{\partial SEE}$ model. Earlier studies e.g. Djamai et al. (2015), Mishra et al. (2018) etc. suggest that the linear model is better suited for dry conditions whereas non-linear models outperform the linear approach under water rich environments (such as Southeast U.S. in this study).

5. One of the assumptions needed for POME is due to non-monotonic behavior of SM. This necessitates the assumption of an inflection point. In the present study, only for 9% of the profiles generated such an assumption was needed. So, although POME is assumption free, the application of POME may not be (still better than LSM regarding number of assumptions). Can the authors reflect on the applicability of POME for other studies? If POME was not applied on SM but instead on the soil water potential (SWP) non-monotonic behaviour across a profile is no longer a concern. This would of course require a relation between SWP and SM, but increasingly assumption free relations are being developed to do so, they only at present take a lot of computational time.

**Response:** We thank the referee for this observation and it is an excellent point. In the dynamic case an additional assumption is required to employ the POME method correctly. This case is clearly identified when the mass balance constraint cannot be met and thus equation 4 has no solution. The assumption is then that the inflexion point is located in the soil layer with the greatest ability to retain water (i.e., field capacity) and thus the SM would be less both above and below that point. In an earlier study Mishra et al. (2015) the authors used a detailed model of soil moisture movement to verify this assumption, and it has generally been verified by the results shown here and elsewhere. As the referee points out, if soil water characteristics are used that are known to be monotonic, then that particular problem is solved, but as stated, assumptions relating actual SM to the variable being modeled (such as SWP) are necessary. So we feel that our approach, based as it is on well known properties of SM, is appropriate at this time.

6. Restricting ALEXI and POME by soil moisture at field capacity means that comparison with in situ data misses out on the field capacity to saturated soil moisture range, especially in the water rich environments of the present study. Can the authors reflect on this?

**Response:** We agree with the reviewer that the relationship between fractional ET and available water content has limitations. It can only effectively provide SM information between wilting point and field capacity. Since ALEXI sense the moisture content of the entire root-zone (~0-100 cm), it is highly unlikely that the entire soil column would experience moisture content greater than field capacity or less than wilting point. Further, if we underestimate conditions above field capacity we may dry out the root zone faster than observed, but theoretically that would be corrected with the next available retrieval. Therefore if ALEXI retrieval is available within next few (~2-4) days the errors would be

7. Looking at Fig 4 the correlation for 6 out of 10 sites decreases with depth. What could cause this? The most dynamic SM can be expected at the soil surface if no preferential flow path are present in the profile. In figure 5 most dynamics can be seen at the 40-100 layer at some sites, indicating preferential flow paths (unless a dynamic shallow groundwater table is present). Some of the sites have a clay type soil which may form cracks and thus provide preferential flow conduits. This might perhaps also explain the decrease in correlation at greater depth, because such flow conduits will only be active under very wet conditions. These may are not accounted for in the POME approach as this is limited to field capacity or is this the result of the overpass time?

**Response:** This is an excellent observation.  As the referee has deduced, there are several issues at work here. First, our simple assumption of the lower boundary condition at 50% of available SM is the main factor causing the decrease in correlation with depth. As the referee states, several sites showed unusually dynamic SM behavior in the lower layers which is not consistent with our assumption. We made this assumption at this point in order to stay within the spirit of the Principle of Maximum Entropy, *i.e.* that a minimum of *a-prior* information should be used in construction of the profile.  However, we note that the lower boundary provides an ideal point to link the profile with model climatology or field conditions if they are known. In this case, because this is somewhat a proof-of-concept study, we did not want to do that and we do note that bias and RMSD generally improved with depth in the POME model. Now, as to the cause of the dynamic behavior noted at certain sites, we have conducted an extensive investigation of this dynamical behavior of SM time series at lower depths and have concluded that this behavior is site specific. Two sites (2013 and 2115) have very porous soils near the surface (2115 even classified as gravel) with clay layers beneath and the other site (2113) is located very near a stream and is probably influenced by surface water intrusion. As the referee states, cracked clay soils could also result in the behavior but we do not think that was the case in these instances. As also noted, the cloud constraints on the ALEXI data impacts its availability and for that reason 3 day composites were employed for both the ALEXI and observed data. So it is unlikely that the ALEXI data availability is the cause of the poor correlation in the lower layers of the SM profile.

8. The conclusions puzzle me regarding the comparison between Noah, POME and SCAN. I have fig 5 and 6 for Noah, and fig 4, 5 and 6 for POME. Why is there only one general Noah/Scan comparison for Noah in fig 6?

**Response:** Thanks for the comment; one of the main purposes of this study is to evaluate the performance of the POME profiles. To that end we used two approaches: one to evaluate the POME profiles against in-situ observations from SCAN sites for ground validations. But there is a scale mismatch issues when comparing gridded product with point observations; second, to avoid scale mismatch issue, we used another gridded SM product from Noah LSM to further evaluate the POME profiles. Here we do not assume that Noah is free from any error, rather its errors are independent of the errors of either POME or in-situ profiles. Through this study, our intension are not to perform an inter-comparison between the three SM products, rather it is to evaluate the POME model performance against two separate and independent data sources. Therefore, we did not focus on the extensive comparison of SCAN with Noah SM product in this study but limited our analysis to general trend as shown in fig. 6.

Specific comments:

Line 44-46: Very confusing line: In general, in-situ SM profile data are only available at a few locations in the US for any given period of time. Why the focus on the US? I am sure globally there will only be a few in situ, but this line somehow seems to suggest there are only US data.
**Response:** We thank the reviewer for pointing this out. We have revised the sentence in the revised manuscript to avoid any confusion regarding data availabilities. (Page-2 lines 15-16)

Equation 2: for consistency please define SMmod as well.
**Response:** The model that defines the relationship between soil moisture and soil evaporative efficiency is defined in the manuscript. Please refer to Page-8 eq. (3).

Table 1: Soil type for site 2037 and 2038 unknown? Also site 2037 typo in Shrubland.
**Response:** We again thank referee for point the type out. We have rectified the type in the revised manuscript.
        As for the soil types for sites 2037 and 2038 (both in SC), the soil information is indeed unavailable on SCAN website for those two sites. Therefore it was intentionally left blank. In the revised MS we have added this information in table caption as well for clarity.

**References:**

[revised manuscript text omitted]

---

## Author Response (AR3)

Refree#2

I have to admit that the authors have made a significant advance in the rewriting of the manuscript. The actual version of the manuscript better highlights the potential of the proposed model. Nevertheless, there is still one major issue that requires some attention in my opinion.

In particular, the approach used to estimate the relative soil moisture in the lower layers is critically linked to the lower limit relative soil moisture, which is a model parameter. This value is a controlling factor that is influenced by the local texture, pedology, hydrogeology and climate. This is not just a random value that one can assign, because is going to affect the model results as it does.

It is clear that this parameter is what we need to know and we need a procedure to identify it. The idea that we can set a value based on experience is certainly not a rigorous approach. Therefore, the message that this paper is making may be misleading for hydrological community.

Even if the authors tried to address my previous comment on this point. I think that the author do not solve the issue. Therefore, I strongly suggest to address this issue in the final version of the manuscript.

**Response**

We appreciate the referee's concern on this point. As mentioned several times in the manuscript, since this was a proof-of-concept study we felt that the use of a constant 50% of available water capacity would be an appropriate lower boundary on the integral and would be in the spirit of the maximum entropy concept of a minimum of *a priori* information available. Since the POME model is applied with effective SM conditions, the 50% parameterization effectively becomes soil independent. Therefore, since soils characteristics do not come into play until the profile is converted to volumetric water content, the issue of soils data availability is not a concern while developing the profiles. We felt that the good results obtained with even this rudimentary assumption highlighted the strength of the entropy approach.  This point is further highlighted in the revised MS Page 10 lines 8-13.

As pointed out in the comment (and by the authors in the manuscript) the actual (volumetric) SM in the lower layers of the column is a function of the soil characteristics and potentially the hydrogeology of the area. In some parts of the world (the U.S. for example) this information might be known and used to set the lower boundary. In fact, as mentioned in one of our earlier responses, this was known in our case but we chose not to use it for the reason given previously. However, in many parts of the world this information would not be known.

That being said, we feel that the referee might be over-emphasizing the importance of the lower boundary. This layer (normally 100-200 cm) is located well below the root mass of most vegetation species (particularly row crops) (Wu et al., 1999) and thus the selection of the lower bound will not affect model results to any great degree. In fact, as the results show, the procedure is most accurate in the middle layers of the soil profile which is where the majority of the root mass is located. Again, we feel that our results confirm this concept since with the exception of the couple of sites where the lower bound oscillated (which has been discussed in an earlier response) our results were very good.

However, we also feel that the selection of the lower boundary is dependent on the use to be made of the resultant SM profile. At this point we feel that the main application of such remotely sensed SM profiles will be to act as proxies of observations, particularly in data limited regions. These profiles can potentially be assimilated into other models of hydrology or agricultural production to produce more accurate and reliable results. Thus we have recommended that the lower boundary is ideally suited to tie the RS soil moisture profile to the model climatology, particularly since it is located below the root mass and thus will exert minimal influence on model results (e.g., ET, primary productivity, biomass, *etc*). We have used the lower bound to tie the RS profile to the climatology of an agricultural crop model in a paper currently under review and have obtained excellent results in estimating crop yields.

Assimilation of remotely sensed SM profiles in a model can be done with effective SM conditions so this will eliminate the profile dependence on underlying soil characteristics during the initial stages. The conversion to volumetric SM must necessarily rely on whatever soils information is in the model to which the RS profiles are to be assimilated.

We apologize that we did not make these points clear (especially potential uses of the procedure) and have revised the manuscript accordingly to reflect the above discussion on pages 2 (Lines 11-18) and 10 (lines 8-13).

[revised manuscript text omitted]

---

## Author Response (AR4)

**Editor Decision: Publish subject to technical corrections** (06 Sep 2018) by Theresa Blume
Comments to the Author:
Dear authors,

I am happy to tell you that your manuscript will be published as soon as you have carried out a minor technical correction: You added this section on page 10 of your revised MS:
"Moreover as discussed earlier, the ultimate purpose of the developed profiles is to be assimilated into a LSM, the lower boundary can serve as a highly effective way to tie the POME profile to the model climatology by using the model SM in the lowest layer of the soil column as the lower boundary on the POME integral. Since this level ( 100-200 cm) is normally well below the root mass of most vegetation species (particularly row crops) (Wu et al., 1999), then its selection will have minimal impact on model results."

In my impression this section is more of a discussion of a potentially different procedure to assign the lower boundary and not something you have actually done for this study and describe the results of in this mansucript. I therefore suggest to move this to the discussion. As you are here talking about two models in the same sentence please also always make it clear which one you are referring to and refrain from simply referring to "model", e.g. at the end of the last sentence.

Thank you for your efforts,
Theresa Blume

**Response:** We agree with the editor, the mentioned section is more suited for discussion than model description. Therefore as per the suggestion, the specified section has been moved to the discussion section, specifically section 5 – 'Error Characterization' page 18 lines 5-9.